# Computational reconstruction of mental representations using human behavior

Laurent Caplette [1] ✉ & Nicholas B. Turk-Browne [1,2]

Revealing how the mind represents information is a longstanding goal of cognitive science. However, there is currently no framework for reconstructing the broad range of mental representations that humans possess. Here, we ask participants to indicate what they perceive in images made of random visual features in a deep neural network. We then infer associations between the semantic features of their responses and the visual features of the images. This allows us to reconstruct the mental representations of multiple visual concepts, both those supplied by participants and other concepts extrapolated from the same semantic space. We validate these reconstructions in separate participants and further generalize our approach to predict behavior for new stimuli and in a new task. Finally, we reconstruct the mental representations of individual observers and of a neural network. This framework enables a large-scale investigation of conceptual representations.

Revealing the information contents of memory is central to elucidating the mechanisms underlying cognition. For example, categorization requires the match of stimulus features to a memorized representation, prediction relies on the transfer of memorized information to sensory areas, and learning updates memorized representations to incorporate new information. Knowing what information is represented is a necessary part of understanding at an algorithmic level how the brain performs a particular task: one cannot know how the mind works without relating specific behaviors to specific stimuli[1–3]. Accessing the content of mental representations is necessary to understand the human mind because it allows us to link sensory features to behaviors. Such knowledge can in turn inform models of human behavior that make specific predictions. For example, understanding which features are part of someone's mental representation of the concept "trustworthy" helps us to predict how they will act in social interactions with different individuals. Characterizing mental representations is not easy, however, as they are not directly observable.

One way to understand the behavior of a black box like this is by probing it with noise[4]. This idea is the basis of the reverse correlation paradigm[5–7]. In the purest form of this paradigm, participants are shown random noise (e.g., pixel noise) on every trial but told that there is a hidden signal on half of the trials (e.g., the letter s) and that they must respond when they think the signal was shown[8]. Averaging the

stimuli associated with these "superstitious" detections reveals the mental representation of that signal, unbiased by external input. This approach has been used to recover representations of letters, facial expressions, and 3D patterns[8–11]. Findings from these and related methods informed and validated models and theories of letter identification, universality of facial expressions, and trustworthiness judgments, among other aspects of cognition[10–13]. One prominent disadvantage of this method however is that it typically requires thousands of trials per observer and is limited to artificial or simple target signals that always have the same pixel representation.

Natural visual categories are defined by abstract features invariant under many linear and nonlinear image transformations, rendering the traditional pixel-based reverse correlation approach inappropriate for capturing their representations. Instead, random sampling of features learned by intermediate layers of convolutional neural networks (CNNs) – CNN features, for short – might be more appropriate for this goal. CNNs are deep neural networks that can be trained on thousands or millions of natural images to learn abstract features that allow them, among other tasks, to categorize objects with a high accuracy[14]. Features from intermediate layers can correspond to object parts or complex textures that map to multiple representations in pixel space (e.g., multiple potential colors, shapes, and orientations)[15,16]. Moreover, these features have been found to predict the brain activity of mid- and

[1]Department of Psychology, Yale University, New Haven, CT, USA. [2]Wu Tsai Institute, Yale University, New Haven, CT, USA.
✉e-mail: laurent.caplette@yale.edu

high-level areas of visual cortex in response to natural images better than previous models[17–19]. Recent neuroimaging experiments have exploited this finding to recover mental representations of natural images[20–24] or to construct superstimuli that maximize the activity of a given brain region[25,26]. By relating CNN features and brain responses, new brain responses can be translated into vectors of CNN feature values (representing how much each CNN feature is associated with these brain responses), which can in turn be used to reconstruct images corresponding to the visual contents of the brain. Existing studies have followed this approach to reconstruct mental representations of stimuli currently available from the external environment. However, recovering internally generated representations has proven more challenging (e.g., with imagined stimuli[24,27].

Of course, CNN features have reduced expressiveness compared to pixels: whereas sampling pixels can allow the reconstruction of any image, sampling CNN features limits the space of possible reconstructions. At the same time, CNN features may be closer to the features represented in the human brain than pixel RGB values (e.g., visual areas are known to be sensitive to combinations of pixels invariant to several transformations rather than individual pixels), and so the subspace of images that can be reconstructed may be more relevant to capturing representations of natural categories. CNN features may not be the most appropriate features, however. Although CNNs can predict human behavior and brain activity well in some tasks with natural images, they struggle when stimuli are altered by experimenters[28–30]. Parameters of 3D generative models may be more appropriate features[28,31]. However, such models are currently restrained to predefined categories (e.g., faces[32–34]). An alternative is the use of adversarially robust CNNs: although imperfect, these networks are more robust to manipulations by experimenters than traditional CNNs and their features seem closer to the ones represented by humans[35,36].

An additional drawback of reverse correlation studies is that they focus on reconstructing a limited number of predefined stimuli[8–11]. A more general, ambitious, and potentially fruitful approach would be to recover a function mapping the labels of all natural visual categories to visual features (see also refs. 37,38). Because both category labels and visual representations can be conceptualized as being positioned in continuous multidimensional spaces, such a mapping might be computable. Moreover, it might be feasible to recover it, or a good approximation, with a reasonable amount of data. Indeed, because multiple labels are semantically related or interchangeable, the semantic space of relevant category labels is relatively low-dimensional[39–41]. Similarly, because multiple images depict the same category, the space of relevant visual representations is also relatively low-dimensional[42–45].

In this study, we aimed to recover and visualize the representations of an arbitrarily large number of complex natural concepts. We focussed on natural concepts because, to have a model of human behavior in the real world, one needs to uncover the representations of useful real-world categories. Most importantly, we aimed to uncover a powerful method that would allow us to reconstruct many representations because this is a step toward a general model of human behavior. Focussing on the representation of a single category would allow us to understand and predict behavior pertaining to that category but may not inform us about other categories. Accessing many representations could allow us to study qualitatively different and more general questions, such as the optimality of representations overall, their dimensionality, and how they cluster together, for example. Finally, we aimed to reconstruct representations in single individuals. A method that achieves this aim would allow us to study inter-individual differences in representations and their origins, for example by relating these differences to distinct developmental, social, or cultural factors.

We developed a method that generalizes reverse correlation in terms of both sensory inputs (stimuli) and behavioral outputs (labels).

First, instead of sampling pixels, we pseudo-randomly sampled the features of an adversarially robust CNN, i.e., the abstract features to which the channels of this CNN respond. The result is a vector of CNN feature values, i.e., a vector of values associated with each feature. These serve as our target feature values for the creation of a stimulus: on each trial, we iteratively optimized an image so that its CNN feature values corresponded to these target feature values. (Note that we refer to this process as sampling CNN features for simplicity, even though we cannot directly sample them and must resort to an iterative optimization procedure; see Methods: CNN-noise stimuli.) The final optimized image is a mix of random CNN features in which no object is clearly visible (CNN-noise stimulus; Fig. 1a). Then, rather than focussing on one or a small set of predefined categories, we asked observers to write any category (between one and three) they perceived in the stimuli, and we transformed their responses to semantic feature values using a pretrained word embedding (Fig. 1b). Such an embedding maps words to features in a continuous multidimensional space according to their semantic content. This experimental paradigm allowed us to relate the entire semantic space of category labels to the visual space of CNN intermediate layer features, and thus to retrieve the visual features associated with any label, even ones that were not supplied during the experiment. From the translated feature values for a given label, we could then synthesize an image approximating the concept's mental representation using a procedure like that used to synthesize stimuli. Our aim for these reconstructions was not to perfectly capture all features of a mental representation but rather to create visualizations that were distinguishable from each other and that were generalizable to other tasks.

We show that our reconstructed mental representations capture meaningful aspects of visual concepts: separate participants recognized what was depicted in the reconstructions and we could predict behavior in a new task and for new stimuli using the representations. Furthermore, these reconstructions differed from those of the neural network used to synthesize the stimuli, showing that simply using the network's representations as a proxy for human representations is not advisable. Finally, we observed idiosyncrasies in the reconstructions across people, which could enable future investigations of individual-level experiences and differences. In sum, we developed an approach that allows conceptual representations in the human mind to be visualized and interrogated.

## Results
### Relating visual and semantic features
Participants provided an average of 2.17 labels per stimulus. After corrections and removal of invalid responses (single characters, numbers, stop words as defined by the NLTK Python library[46], and words unrecognized by the word embedding), this corresponded to 2578 unique words, of which 369 were provided at least 10 times each (Fig. 2a; Fig. S5; Table S1). The most frequent words were grass (607 responses), sky (592), tree (450), dog (355) and bird (355). Although most responses were objects or concrete things, there were some more abstract concepts. Looking at the 350 validated labels (labels named 10 times or more and labels from the Visual Genome database; see below), four labels (1.1%) were not nouns (green, white, black, and dark) and an additional eight (2.3%) were clearly not basic-level (animal, building, clothes, furniture, fabric, buildings, container, and vehicle). All words were included in the analyses.

We transformed each word into semantic feature values using a pretrained word embedding, averaged these vectors of feature values into a single one per stimulus, and reduced the dimensionality of these average semantic feature values using principal components analysis (PCA). The resulting principal components (PCs) are reduced representations of the semantic content of the stimuli as perceived by the observers. We also used PCA to reduce the dimensionality of the vectors of CNN feature values associated with the stimuli. Finally, we

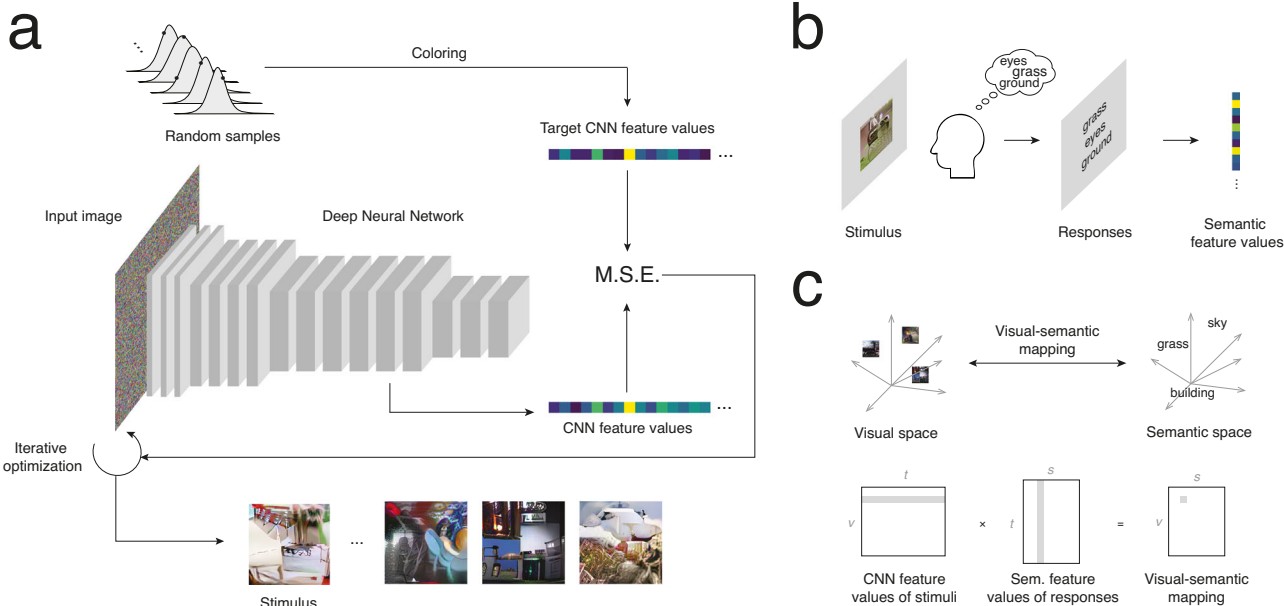

**Fig. 1 | Experimental methods and analyses. a** Illustration of the stimulus synthesis procedure. Random samples are drawn from uncorrelated distributions. A coloring (inverse whitening) transform is applied to cast these samples to the original CNN feature space (these are the target CNN feature values). An image is iteratively optimized from noise so that its actual CNN feature values are similar to these target CNN feature values. The result is referred to as a "CNN-noise" stimulus (i.e., a stimulus whose CNN feature values are pseudo-random) and will be used in the experiment. Some stimulus examples are shown at the bottom of the panel. M.S.E. = Mean squared error. **b** Experimental paradigm. On each trial, a CNN-noise stimulus is shown to a participant for 5 s. The participant then writes 1–3 labels indicating what was perceived in the stimulus. These labels are transformed into a vector of semantic feature values summarizing the semantic content of the stimulus as perceived by the observer. **c** Overview of the analyses. Visual representations and CNN-noise stimuli lie in a high-dimensional space of CNN features (visual space). Label responses lie in a high-dimensional space of semantic features (semantic space). We can infer a mapping between these spaces by taking the matrix product of the v (CNN visual features) × t (trials) matrix of stimuli and the t × s (semantic features) matrix of responses. The result is a v × s visual-semantic matrix indicating how each CNN feature is related to each semantic feature. Gray shaded areas indicate an example CNN feature and its values across trials (left), an example semantic feature and its values across trials (middle), and their association inferred by the visual-semantic matrix (right).

inferred the linear associations between the semantic PCs and the CNN PCs (Fig. 1c). The result is a visual-semantic matrix indicating how much each CNN PC is related to each semantic PC (Fig. 2b). There were 67 significant associations (Z < −4.76 or Z > 4.75; $p < 0.05$, two-tailed, FWER-corrected): 21 unique CNN PCs were positively and negatively associated to 14 unique semantic PCs. The most important semantic and CNN PCs, along with their associations, are visualized in Fig. 2c. For example, a semantic PC associated with nature concepts (PC #2) is strongly correlated with CNN PCs that partly code for grass-like and water-like textures (PCs #1, #4 and #7). Similarly, a semantic PC associated with humans/animals (PC #3) is related to CNN PCs seemingly representing skin textures, fur textures, and animal faces (PCs #2, #4 and #5). Our visual-semantic matrix further allows us to uncover and visualize the words most associated with a given CNN PC, improving the interpretation of features (Fig. 2d). This specific visualization method could be applied to units of different neural networks to aid explainability[47] and reduce observer bias.

**Reconstructing mental representations of visual concepts**

Using the visual-semantic matrix we inferred, we can now retrieve the CNN feature values associated with any concept. Specifically, we first retrieve the semantic feature values associated with a specific concept label using the word embedding and we use our visual-semantic matrix to retrieve the associated CNN feature values. Importantly, these values can then be supplied for reconstruction, with the resulting image being an approximate visualization of the mental representation of the concept (successful reconstructions and failure cases are seen on Figs. 3a, b, respectively).

We validated the reconstructed representations of 350 words in an additional behavioral experiment: the 250 words most-named in the experiment plus the top 100 most frequent words from the Visual Genome database[48] labeled less than 10 times by participants (and thus not part of the 250 most-named). On each trial, 50 participants were shown a reconstruction and had to choose between two labels: the true label and an incorrect label randomly chosen among the labels of the other reconstructions[24]. Note that because the non-matching label was chosen randomly and that there were few non-basic-level category labels, nearly all comparisons were between two basic-level category labels. The reconstructions for the 250 most-named words were recognized well on average, with a mean accuracy of 88% (significantly above chance, $p < 0.001$, one-tailed). For the 100 additional frequent concepts from the Visual Genome database, the mean accuracy was 74% ($p < 0.001$, one-tailed). When considering all validated labels in the Visual Genome database, irrespective of whether they were named more or less than 10 times during the experiment (209 concepts), the mean accuracy was 84% ($p < 0.001$, one-tailed). Overall, 270 concepts (out of the 350 that were validated) were individually recognized significantly above chance (accuracy >75%; $p < 0.05$, one-tailed; Fig. 3c). This proportion of significant concepts was higher than would be expected by chance (random resampling of participants to create an empirical null distribution, $p < 0.05$, one-tailed, FWER-corrected). Within the 100 concepts named less than 10 times by participants, 47 were individually significant. Among best recognized concepts were bird, building, and people, with accuracies of 100%. Worst recognized were jeans, white, and feet (38%, 30%, and 25%, respectively).

We also performed an additional, more stringent, validation task in which a new set of 50 participants spontaneously labeled 100 reconstructions. Specifically, each participant was shown the reconstructions of the 100 most-named concepts and had to provide 3 potential labels for each reconstruction. For each reconstruction,

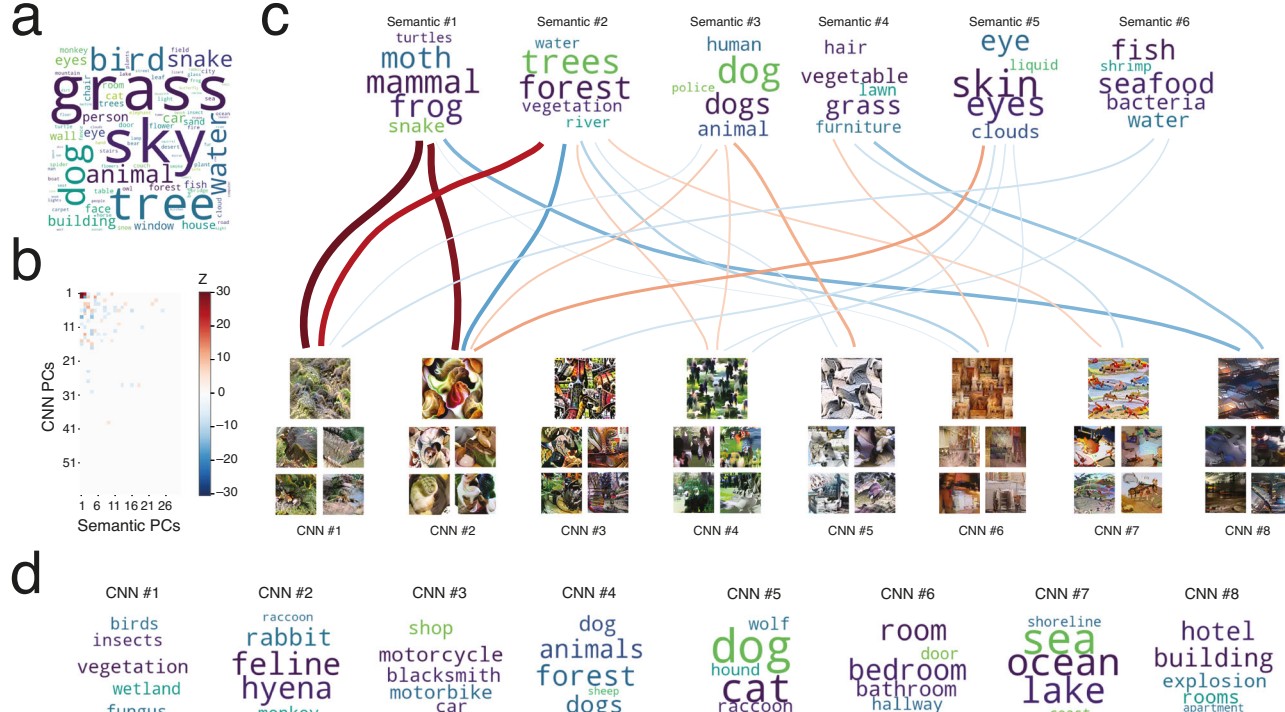

**Fig. 2 | Relating visual and semantic features. a** The 100 most common responses provided by participants. The size of the word is approximately proportional to its frequency. **b** Illustration of the significant associations between the first 60 CNN principal components (PCs) and the first 30 semantic PCs ($p < 0.05$, randomization test, two-tailed, FWER-corrected). **c** Detail of the significant associations between the first 6 semantic PCs and the first 8 CNN PCs. The color of the curves represents the strength and polarity of the association (see scale in (**b**)), and the thickness is proportional to its absolute value. Semantic PCs are summarized by the 5 words provided in the experiment that loaded most strongly (word size approximately proportional to closeness). CNN PCs are summarized by both a synthesized image maximizing the value of that PC (top) and the 4 stimuli that loaded most strongly (bottom). **d** Summary of the semantic content associated with the top 8 CNN PCs. The 5 words that corresponded most closely to the semantic feature values associated with each CNN PC are shown (word size approximately proportional to closeness).

participants wrote many different labels. On average, there were 68.9 unique labels per concept (standard deviation across concepts = 16.3). Despite the greater difficulty of this task, several concepts were recognized with high accuracy: the most common written label was the correct label for 37 of these concepts (significantly above the 1.1 concepts that would be labeled correctly on average by chance; tested with random permutations of participant responses across reconstructions, $p < 0.001$, one-tailed; 95% C.I. = 28–47). Results were consistent with the first validation study: all 37 concepts were among those successfully recognized in that study. The most successful concepts included bird, tree, and people (correctly answered by 49, 47, and 46 participants, respectively). This test relies on writing the exact correct label, however. To account for participants writing inexact but closely related labels, we analyzed the semantic features of the responses and how similar they were to those of the true labels. Strikingly, we found that responses that were close to the true label in semantic space were more frequent than responses that were far in semantic space for 85 of the 100 validated concepts (inverse linear relationship between response semantic distance and frequency, significance tests of the slope coefficients, all $t(150) < −3.68$, $d_z > 0.30$, $p < 0.05$, one-tailed, normality assumed, FWER-corrected).

It was possible to reconstruct many closely related concepts. A visual inspection shows that subtle differences in visual features between these can apparently still be revealed, although additional validations would be necessary to conclude this with certainty (Fig. 4). This also seems to be true for singular and plural forms of the same concept, with more smaller repetitions of the concept across the image when the plural form of a given word was input (Fig. 4f). Note that 270 does not seem to be the upper bound of the number of

concepts that can be reconstructed: other words can be input that result in seemingly successful reconstructions (e.g., pond, bushes and shark on Fig. 4) although, again, we cannot know for sure without validating them too.

We conducted several control analyses. First, to verify that there was no bias toward the reconstructed concepts inherent in our reconstruction method, we reconstructed concepts after randomly permuting the vectors of semantic features across trials (disrupting visual-semantic associations). As expected, the reconstructions were greatly altered by this shuffling and they varied widely depending on the specific permutation, showing that our results are not an artifact of the method (Fig. 5a). To quantify this and further validate our method, we recruited another group of 25 participants to label some of these *null* reconstructions, in addition to real ones. Specifically, four images were chosen for each concept: the real reconstruction and three random null reconstructions associated with the same concept. Because of the larger total number of stimuli for this experiment, 45 concepts were randomly chosen among the 100 most-named concepts and participants could provide one label for each image. The correct label was given to the real reconstruction more often than to any associated null reconstruction for 78% of concepts (35/45 concepts; significantly above chance, $p < 0.001$, one-tailed; 95% C.I. = 29–40). When analyzing responses as vectors in a continuous semantic space (to account for the fact that some responses are more similar than others), the responses given to real reconstructions were semantically closer to the correct label than the responses given to associated null reconstructions for 98% of concepts (44/45 concepts; significantly above chance, $p < 0.001$, one-tailed; 95% C.I. = 42–45). In addition, responses were more consistent (i.e., lower entropy of the response probability

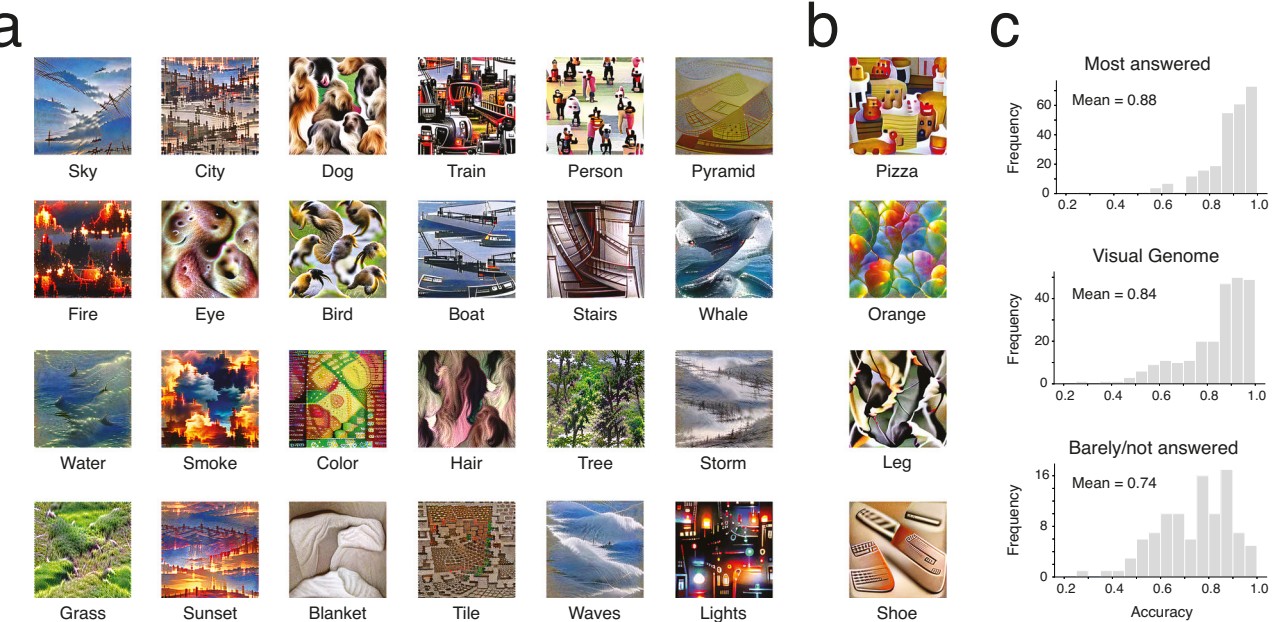

**Fig. 3 | Reconstructions of mental representations and results of the first validation study.** Note that the target concept may be tiled across the image. **a** Examples of successfully validated reconstructions (accuracy >75%; $p < 0.05$, randomization test, one-tailed, FWER-corrected.) **b** Examples of failed reconstructions. **c** Histograms of recognition accuracies for concepts most named in the main experiment (top), concepts present in the Visual Genome image database (middle), and concepts from the Visual Genome database answered less than 10 times in the main experiment (bottom) (all mean accuracies significant, $p < 0.001$, randomization tests, one-tailed, FWER-corrected).

distribution) for real reconstructions than any associated null reconstruction for 80% of concepts (36/45 concepts; significantly above chance, $p < 0.001$, one-tailed; 95% C.I. = 31–41), and their variability in semantic space (i.e., the trace of the covariance matrix) was smaller for real reconstructions than any associated null reconstruction for 89% of concepts (40/45 concepts; significantly above chance, $p < 0.001$, one-tailed; 95% C.I. = 35–44), suggesting that real reconstructions contained more readily perceived meaning.

We then reconstructed images from the same CNN feature values using different seeds, to visualize the impact of the random seed used for reconstruction. Although slight differences can be observed, these concern mostly the location of the features across the image (Fig. 5b). Following this, we visualized the uncertainty associated with the CNN feature values. To do so, we obtained the lower and upper bounds of 95% confidence intervals around the CNN feature values and reconstructed images from these. The true reconstruction is likely to fall between these bounds and so these images give more information about the true representation than the reconstruction from the observed feature values alone (Fig. 5c).

Finally, we analyzed the impact of the semantic embedding on reconstructions. We first assessed whether we could reconstruct the representation of a concept without using the concept's name, by relying on other words to learn the visual-semantic matrix. For each reconstruction, we removed all responses that contained the concept's name and re-ran all analyses. The resulting reconstructions were quite similar to the originals (Fig. 5d), even for the 10 most-named words (correlations of 0.63–0.92 between the CNN features from both analyses for the top 10 words; $Z = 3.29$–4.16; all $p < 0.001$, one-tailed, FWER-corrected; mean $r = 0.85$, 95% C.I. = 0.840–0.852). This performance is notable given that we expected these most-named concepts to be most affected by this analysis (because many of the original responses that contributed to their reconstructions were removed), in contrast to less-named concepts for which fewer responses were removed by definition. We then repeated the analyses without using a semantic embedding at all (Fig. 5e). Specifically, for each

representation to reconstruct, we repeated the analysis while replacing the semantic feature values of a stimulus with a 0 or 1 indicating whether the concept was labeled on that trial (see Methods: Investigation of the effect of the semantic embedding). This analysis was feasible only for words that were named frequently. We postulated that, for these frequent words, the representation uncovered by this analysis would closely approximate the true representation. The representations uncovered by our main analysis thus need to be close to these representations. Indeed, reconstructions were similar and CNN feature values for the 10 most-named concepts were strongly correlated with those obtained for these concepts in the main analysis (correlations of 0.91–0.99; $Z = 3.75$–5.16; all $p < 0.001$, one-tailed, FWER-corrected; mean $r = 0.96$, 95% C.I. = 0.961–0.964; Fig. 5e). Correlations declined as label frequency decreased (Log frequency explains 64% of the variance in Fisher-transformed correlation over all words named 10 times or more; 95% C.I. = 0.633–0.652; $p < 0.001$, one-tailed). One advantage of the semantic embedding is to extend beyond these frequent cases to reconstruct representations of less common labels, or even labels not generated in the experiment.

## Predicting semantic content and similarity judgments

We then tested whether our visual-semantic matrix could predict behavior in contexts other than our experiment. Such analyses allow us to infer whether the matrix is general, i.e., representative of the population and applicable to other situations. We first verified whether it could predict the semantic content of new stimuli. To do so, we computed the semantic feature values associated with the five practice stimuli that all observers saw (and thus for which the semantic content could be estimated reliably) and that were not used to create the matrix. We then tried to predict these feature values based only on the visual content of the images, using our visual-semantic matrix (Fig. 6a). We obtained an average cosine similarity of 0.30 (95% C.I. = 0.17–0.41; $Z = 20.13$; $p < 0.001$, one-tailed; individual cosine similarities ranged from 0.09 to 0.49) between the predicted and true semantic feature values, indicating that our matrix can generalize to new stimuli.

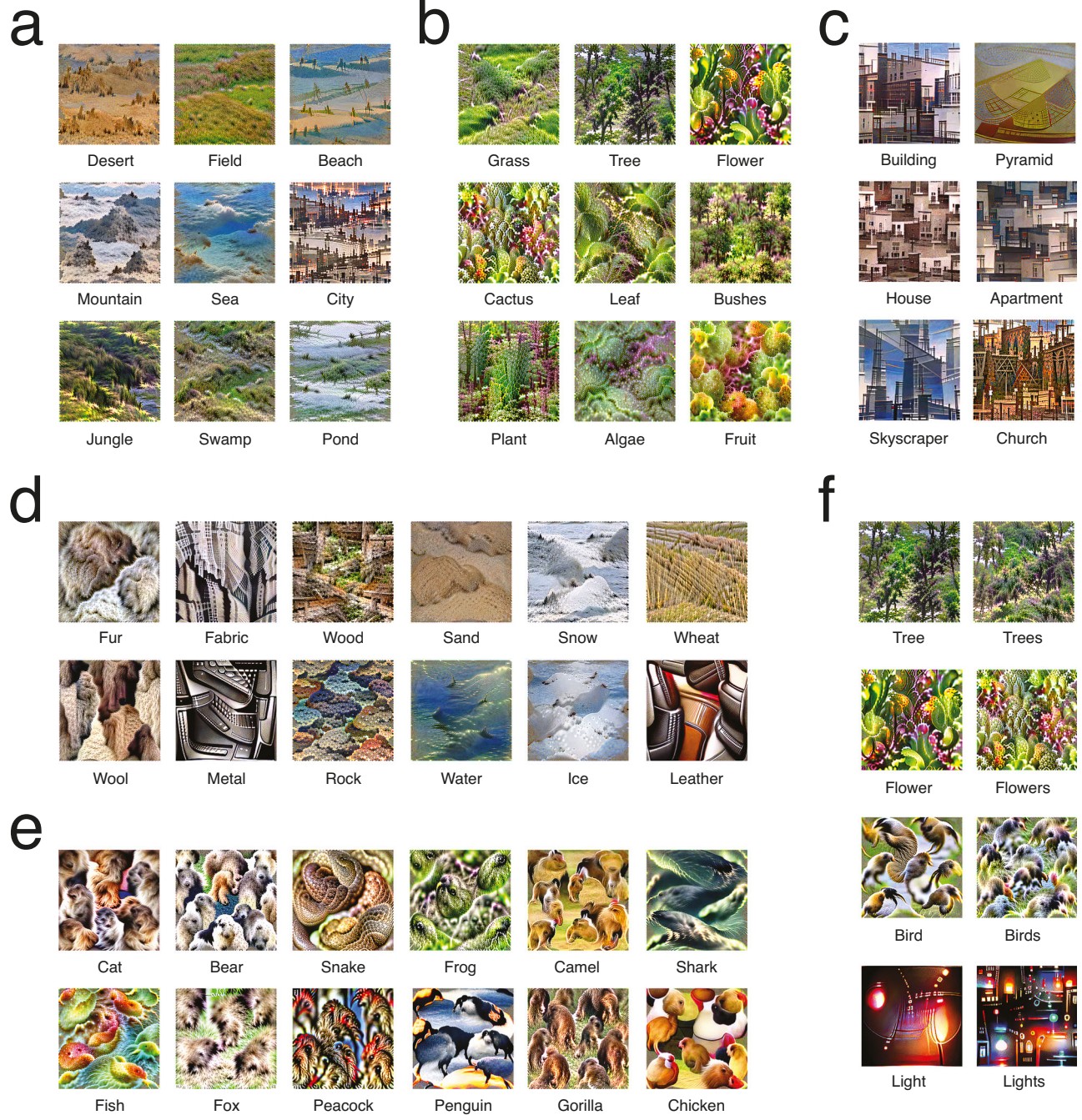

**Fig. 4 | Reconstructions of specific types of concepts.** Note that the target concept may be tiled across the image. **a** Outdoor scenes. **b** Vegetation. **c** Buildings. **d** Materials and elements. **e** Animals. **f** Singular/plural concept pairs.

We also assessed whether we could predict the stimuli that have been classified by participants as depicting a specific concept. To avoid circularity[49], we used the set of stimuli and responses from the individual representations experiment (see below). For each concept, we predicted the stimuli most likely to have been classified as containing the concept and compared these predictions to ground truth (Fig. 6b). When doing this with the 10 most-named concepts, our predictions were better than chance for 9 of 10 concepts (all except eyes; Dice coefficients = 0.19–0.64; Z = 2.16–11.39; p ranged from <0.001 to 0.15, one-tailed, FWER-corrected). Even stimuli predicted as containing a concept but for which the concept was not labeled seemed to include it in many cases (e.g., see incorrect predictions for grass, sky, tree, and water in Fig. 6b). This suggests that some of the prediction errors

reflect a misestimation of the semantic content of the stimuli (because of too few participants/responses) rather than an inaccurate visual-semantic matrix. Overall, this result indicates that our visual-semantic matrix can be used to infer the semantic content perceived in new images.

Finally, we wanted to explore whether our matrix could predict human behavior in other tasks. To do so, we verified whether the similarity structure of the visual representations uncovered with our visual-semantic matrix predicts how independent observers judge similarity between concepts (Fig. 6c). Participants placed written visual concepts on a 2D plane according to their semantic similarity (data from ref. 50). Even though participants were instructed to use semantic information to make their judgments, we predicted that they

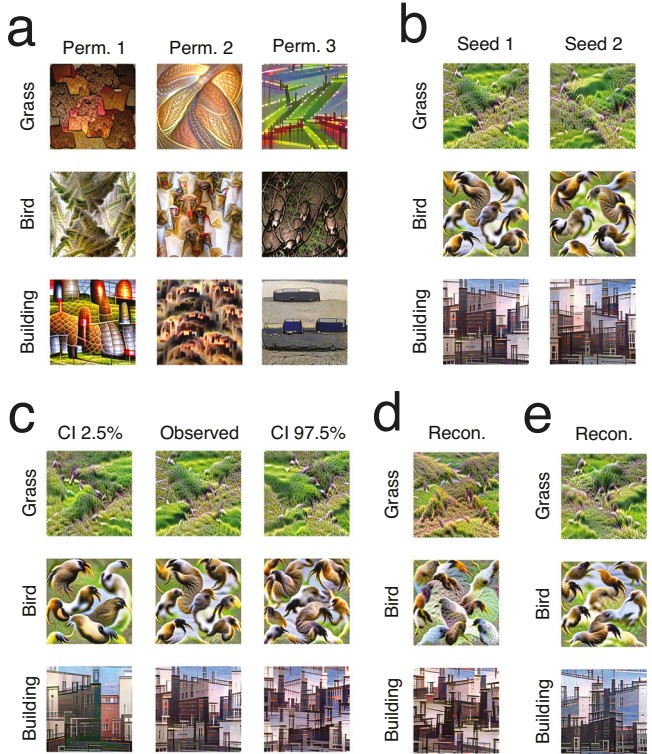

**Fig. 5 | Control reconstructions and analyses, for 3 of the 10 most-named concepts. a** Reconstructions using visual-semantic matrices from a null distribution. **b** Reconstructions from two different random seeds. **c** Reconstructions from the lower and upper bounds of 95% confidence intervals (C.I.) around the observed CNN feature vectors. **d** Reconstructions using visual-semantic matrices created without the reconstructed concept's labels. Correlations between the feature vectors from this analysis and the main analysis for the three depicted concepts are 0.85, 0.87, and 0.92, respectively. **e** Reconstructions using no semantic embedding. Correlations between the feature vectors from this analysis and the main analysis for the three depicted concepts are 0.99, 0.97, and 0.97, respectively.

would still use visual information and that we might capture a unique part of the total variance. Indeed, the visual representations RDM correlated strongly with the similarity judgments RDM ($\rho = 0.56$; 95% C.I. = 0.458–0.647; $Z = 20.04$; $p < 0.001$, one-tailed) and this correlation was stronger than with the RDM based on semantic representations ($\rho = 0.48$; 95% C.I. = 0.357–0.591; significantly weaker, $Z = 2.41$; $p = 0.024$, two-tailed), indicating that visual similarities better explain semantic similarity judgments than semantic embeddings trained on word co-occurrences. Together, these results suggest that representations uncovered with our visual-semantic matrix generalize beyond our observers, task, and stimuli.

**Investigating the representations of the deep neural network**
We then aimed to determine whether human representations were different from the representations of the neural network used for image synthesis. This is important for several reasons. First, uncovering differences between the representations of humans and of a deep neural network (DNN) would show the added value of our method and that using a DNN's representations as a proxy for human representations is insufficient. Moreover, it would reveal that our method is a useful tool to analyze representations of artificial neural networks, in addition to human representations, and potentially to compare representations of different DNNs to each other (to identify the ones that capture behavior better). To achieve this, we repeated the experiment but used as the responses for each stimulus the labels of the three classes (out of the 1000 ImageNet classes) that the network estimated had the highest probability of being depicted in that

stimulus. We reconstructed the network's representations in the same way as we did for the human representations. Resulting reconstructions often look superficially similar to reconstructions of human representations but with the concept made less clear or even unidentifiable (Fig. 7a). We then analyzed whether representations were significantly different between groups (humans vs. DNN) while accounting for different noise levels. Specifically, we projected the visual-semantic matrices of both humans (Fig. 7b) and the network (Fig. 7c) to a common semantic space, divided the data in halves, and compared the features of representations within and across groups. Visual features were more correlated within group ($r_{within, DNN} = 0.85$; $r_{within, human} = 0.56$; $r_{between} = 0.49$ and 0.46; $r_{within, mean} = 0.70$ vs $r_{between, mean} = 0.48$; 95% C.I. = 0.701–0.705 vs 0.477–0.483; $Z = 8.73$; $p < 0.002$, two-tailed), suggesting limits in the correspondence between DNNs and humans (see also, e.g., ref. [51]).

**Revealing representations of single individuals**
Interindividual differences in mental representations have been documented in several cases[11,52,53]. Being able to reveal such differences is a strength of methods using human data (behavioral, as in reverse correlation experiments, or neural, as in brain reconstruction paradigms). Using a neural network or other model as a direct way to generate images, despite having tremendous value, will never capture such representational idiosyncrasies. Indeed, a single model cannot capture the subtle biases and particularities of the representations of specific individuals. To assess whether our method can reveal how conceptual representations vary across individuals, we asked 8 observers to each perform 750 trials of the experiment over the course of several days (all observers saw the same stimuli). We reconstructed representations as before but now for each individual. For the most frequently labeled concepts, reconstructions largely succeeded despite visibly higher levels of noise (Fig. 8a). When visualizing an approximation of the structure of all representations on a 2D plane, representations of a given concept for different participants were clustered (Fig. 8b). Nevertheless, there were differences, as representations for some concepts were more scattered than others (e.g., water, field, and sky in Fig. 8b). To assess whether representations were truly individually unique and stable, we performed an analysis like the one performed to compare human and CNN representations, comparing within-individual to between-individual correlations of CNN feature values. CNN feature values were more similar within individual ($r = 0.22$ vs 0.11; 95% C.I. = 0.195–0.243 vs 0.104–0.124; $Z = 15.00$, $p < 0.002$, two-tailed), creating the possiblity of using this approach to characterize individual differences in visual concepts and semantic memory. Future research should investigate these representational differences, their origins, and their impact on behavior (e.g., accuracy and response times in object recognition tasks) in more detail.

## Discussion
In this study, we reconstructed the contents of mental representations of complex natural categories from purely behavioral responses. Specifically, we synthesized images that were random mixes of abstract visual features, mapped these features to visual categories, and visualized mental representations of these categories. Our method allowed us to successfully reconstruct the representations of several concepts. Seventy percent (270/350) were recognized above chance against other reconstructions in a 2AFC task and 37% (37/100) were accurately labeled in an open-ended labeling task. When analyzing the semantic content of labels, 85% (85/100) were well recognized, with semantically related responses being more frequent than semantically distant responses. We were able to recover the representations of diverse concepts, spanning multiple domains such as animals, vegetation, buildings, colors, materials, and objects. Importantly, it might be possible to successfully reconstruct additional concepts; we could only validate a finite subset of all possible representations.

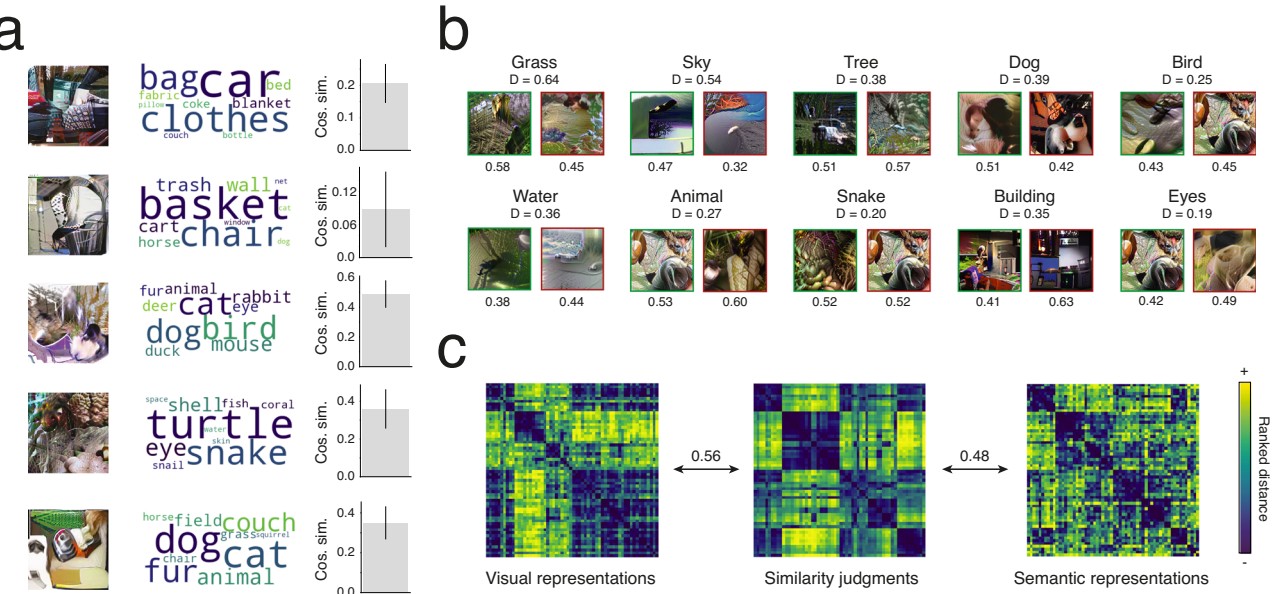

**Fig. 6 | Prediction analyses. a** Prediction of the semantic content of held-out stimuli. Each predicted stimulus is illustrated (left) with associated responses (middle) and the cosine similarity between the true and predicted semantic feature values (right). The average cosine similarity is 0.30 (95% C.I. = 0.17–0.41; Z = 20.13; $p < 0.001$, randomization test, one-tailed) and individual cosine similarities ranged from 0.09 to 0.49. Error bars reflect standard errors of bootstrap distributions (not of samples of individual participants). **b** Prediction of stimuli associated with the 10 most-named concepts. For each concept, stimuli correctly (green frame, left) and incorrectly (red frame, right) predicted as illustrating the concept are depicted. In each case, the depicted stimulus is the one predicted as most likely to include the concept (i.e., whose CNN feature values are most correlated to the CNN feature values of the concept; this correlation is indicated below each stimulus). For each concept, the Dice coefficient, representing the accuracy of the prediction for that

concept overall, is below the name of the concept. All Dice coefficients are significant except for "eyes" (0.19–0.64; Z = 2.16–11.39; p ranged from <0.001 to 0.15, randomization tests, one-tailed, FWER-corrected). **c** Prediction of similarity judgments. The rank-transformed distances between the concepts according to their CNN feature values derived with our visual-semantic matrix (left), to behavioral similarity judgments (middle), and to their semantic feature values derived with the semantic embedding (right). Second-order correlations between the behavioral judgments distance matrix and the other similarity matrices are indicated between the matrices. The visual representations RDM correlated strongly with the similarity judgments RDM ($\rho = 0.56$; 95% C.I. = 0.458–0.647; Z = 20.04; $p < 0.001$, randomization test, one-tailed) and this correlation was stronger than with the RDM based on semantic representations ($\rho = 0.48$; 95% C.I. = 0.357–0.591; significantly weaker, Z = 2.41; $p = 0.024$, randomization test, two-tailed).

In addition, we were also successful in predicting the semantic content of held-out stimuli and behavioral similarity judgments from separate observers. These results suggest that our visual-semantic matrix generalizes beyond the confines of our experimental data to new stimuli, observers and tasks. Furthermore, we could recover representations of the neural network used in this study and of individual observers. We observed that the representations of the CNN were significantly different from those of human observers. This suggests that simply using the CNN's representations as a direct proxy for human representations is not sufficient. In addition to modeling the human representations more accurately, our method could also be used to quantify the differences in conceptual representations between different neural networks or other models. We also showed that our method can be used to assess representational differences across individuals, something that a single general model cannot do. Future studies should characterize these representational differences more fully and assess their impact on behavior, for example, whether these idiosyncrasies affect accuracy or response time in categorization or discrimination tasks. The developmental and sociocultural origins of these idiosyncrasies would also be interesting to investigate.

It required an average of 37 trials (or 80 responses) to reconstruct the representation of a concept so that it can be recognized above chance when pitted against other reconstructions. This is a major improvement over the 20,000 trials required to retrieve the mental representation of a static letter s using pixel noise[8]. Recovering so many representations with so few trials was possible because of several methodological choices. First, allowing participants to give multiple responses on each trial resulted in multiple data points per stimulus. The cost in time for this increased information was smaller than

increasing the number of stimuli similarly (especially given automatic suggestions of word completions that may have accelerated response entry for slow typers). Second, using CNN features instead of pixels as features dramatically constrained the search space of our experiment. Not only is this feature space lower-dimensional than the images (1024 channels vs. 50,176 pixels), but it is more proximal to the concepts we sought to reconstruct. The features represent object parts or complex textures that are closer to the natural visual categories than individual pixels. Of course, the ultimate goal is then to recreate a pixel image from these CNN features, but this synthesis process is completely separate from the behavioral experiment itself and it does not rely on any human data. As a corollary, the representations of different concepts are likely more separable in that space[54,55]. Third, treating each response word as part of a continuous multidimensional space allowed us to exploit relationships between words without needing additional trials to retrieve the representation of each word. Indeed, in our analyses, visual features were associated with dimensions of the multidimensional semantic space (i.e., semantic features) rather than specific words. Thus, to reconstruct the representation of one concept, we benefitted not only from trials in which that concept's name was reported, but from all trials. Fourth, we further reduced the dimensionality of both the sampled visual space and the semantic response space by performing principal component analyses.

Our paradigm is based on the idea of reverse correlation[5,6] in which some features are randomly sampled on every trial. This allows us to uncover the features associated with a target concept in an unbiased way. If we were to show natural images of concepts instead of random features, our reconstructed representations would be biased by this set of natural images. There are internal biases in our

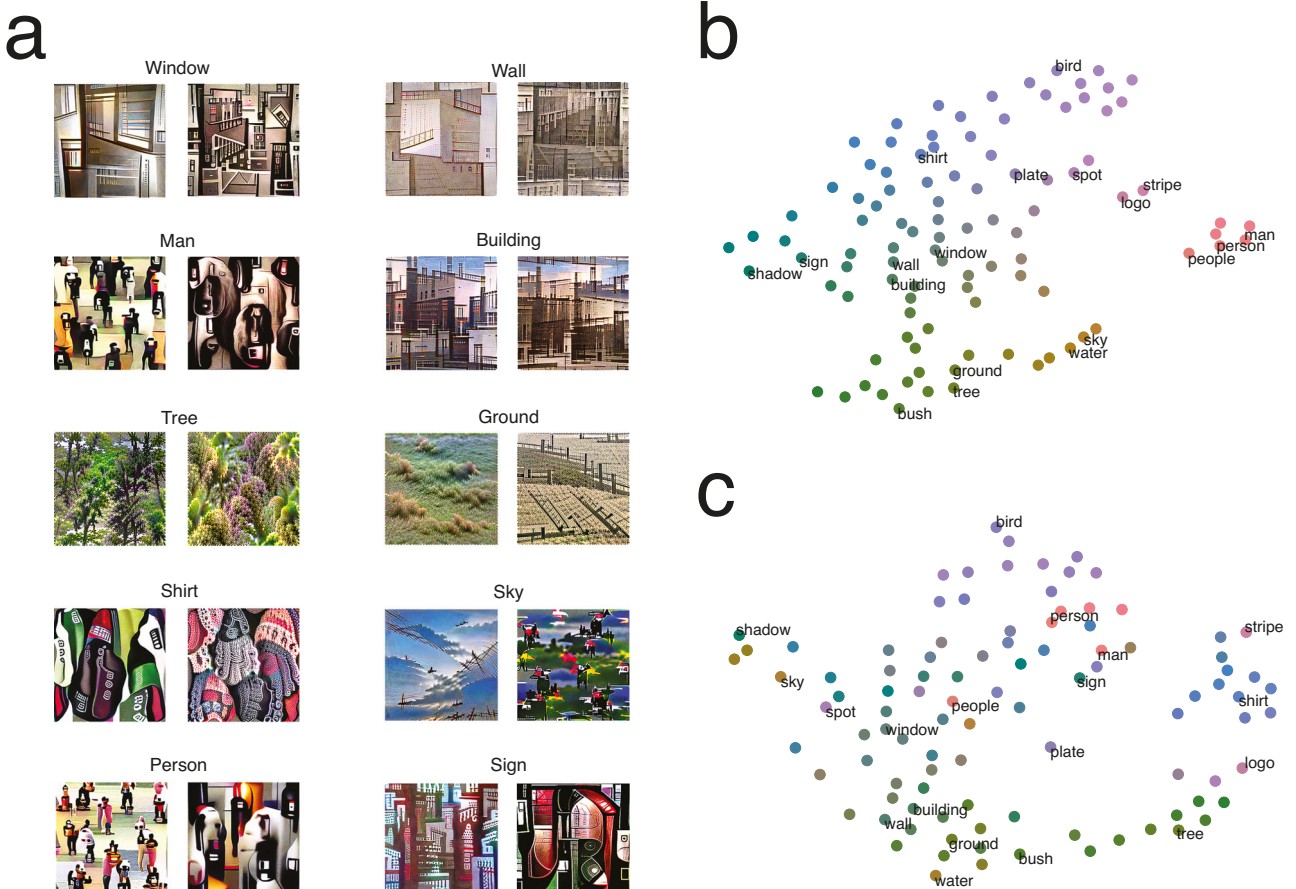

**Fig. 7 | Investigating the representations of the convolutional neural network.** **a** Reconstructions of human (left) and CNN (right) representations for the top 10 concepts from the Visual Genome database. **b** t-SNE plot summarizing the structure of the human representations of the top 100 concepts from the Visual Genome image database. Labels highlight the 10 concepts depicted in panel A and selected additional concepts. Colors gradually change in both axes. **c** Same as panel (**b**) for the CNN representations. The concepts highlighted with labels and the associations between colors and concepts match panel (**b**), allowing for comparison of relative positions. This plot was aligned as best as possible with the plot in (**b**) using a Procrustes transformation. Overall, CNN feature values were more correlated within group than between groups (humans vs CNN; r = 0.70 vs 0.48; 95% C.I. = 0.701–0.705 vs 0.477–0.483; Z = 8.73; p < 0.002, randomization test, two-tailed), indicating that representations were different.

representations that are not present in natural images: we typically focus on (and therefore remember) some features more than others (e.g., ref. 56). Using natural images would also prevent us from uncovering interindividual differences in representations, because most natural images would be labeled the same way by all observers. Interindividual differences may be mostly visible in the distinct weighting of specific features, and they would likely not be uncovered if the images shown comprised all relevant features. Our CNN-noise stimuli force the participants to rely on subtle features that may be perceived differently across observers and allow us to uncover the edges of the representations that better reveal individual variation (in contrast to their centers, likely to be agreed upon by everyone). Finally, such an approach would not allow us to recover representations of concepts with no associated natural images, such as dragon or ghost or strange. In sum, using random features allowed us to recover good reconstructions of multiple concepts because we could sample the relevant feature space in an exhaustive and unbiased way.

We must note however that our method is distinct from reverse correlation in some ways. Most importantly, participants do not have to hold a specific representation in their working memory while viewing the stimuli, before deciding whether the stimulus corresponds to their representation or not. This would be more difficult for participants to achieve with complex non-pictorial representations such as the ones we are targeting. Rather, we ask that they label each stimulus according to what they perceive most prominently in them. We assume that participants have abstract representations of multiple concepts encoded in their mind and that, to label the stimulus with a certain concept's label, they have perceived in it features that correspond (at least partially) to their representation of that concept. However, because of this, the features that we then retrieve with our analyses might be incomplete or indirect reflections of the mental representation.

Note also that conceptual representations live in a more abstract space than the image space, with multiple images linked to a given concept. Thus, our reconstructions are approximate visualizations— i.e., projections onto the image plane—of mental representations. Specifically, our reconstructed image for a concept visualizes the CNN features most strongly associated with that concept. Indeed, the reconstruction is optimized from the recovered feature values of the category, indicating how much each CNN feature is associated with the concept. If, for example, leaves are strongly associated to the concept tree but less so to pinecones, the reconstruction will depict a deciduous tree rather than a conifer, although both are part of the concept tree in the minds of the observers.

Despite its success and potential, there are some limits to our method. Notably, whereas sampling pixels involves directly varying the value of the elements that constitute the image, sampling abstract CNN features requires choices about how to design the features that

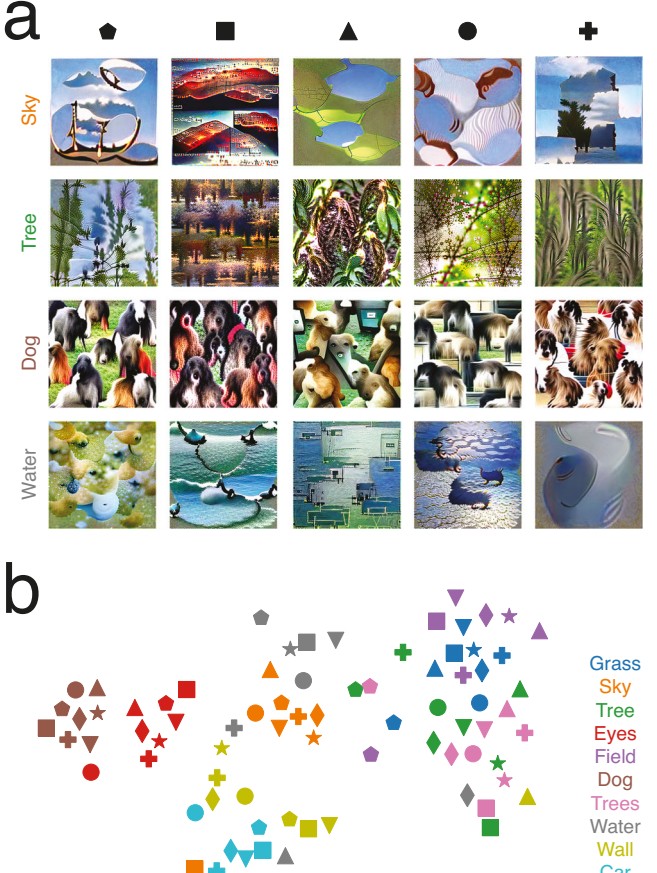

**Fig. 8 | Reconstructing individual representations. a** Reconstructions of selected concepts for five participants. Shape symbols and word colors refer to panel (**b**) **b** t-SNE plot summarizing the representational structure of the 10 most-named concepts (colors; see legend on the right) for all 8 participants (shapes). Overall, CNN feature values were more similar within individual than between individuals (r = 0.22 vs 0.11; 95% C.I. = 0.195–0.243 vs 0.104–0.124; Z = 15.00, *p* < 0.002, randomization test, two-tailed), indicating that representations were individually unique.

features of natural images predict the brain activity of participants viewing these images with good accuracy[17–19,42,60], suggesting some correspondence between them and the features represented by the brain (although see ref. [28,31]). However, the layer must be chosen with care: it must not be too close to whole objects, to allow for the discovery of minimally biased representations, and not too close to raw pixels, to be more evocative of categories and allow for a more efficient search for representations. In future iterations of this paradigm, other layers could be chosen, or different layers could be combined across trials or even within trial.

Layers of other CNNs trained for object categorization (e.g., the more recent FixEfficientNet[61] or Vision Transformer[62,63]) or of deep neural networks trained for other objectives, such as generative adversarial networks (GANs)[24,64], could also be used as the feature space. In addition, using a network trained for scene categorization might allow us to sample more complex multi-object features if we use a relatively high layer. Training the networks on datasets that more closely resemble the distribution of natural categories relevant to humans might further improve the results[65]. Neural networks trained on biased datasets such as ImageNet (with an overrepresentation of dogs and other categories) are likely to bias the features that are sampled, in turn influencing the categories answered in an open labeling task such as ours. Although this does not alter the set of categories that can be reconstructed with our method (since any concept label recognized by the word embedding can be input), categories labeled more frequently might be reconstructed more successfully. In general, the specific architecture, training task, training stimuli, and learning rule are all factors that may alter the features sampled and influence our results.

Some authors have questioned the idea that any deep neural network trained on 2D images can be a good model of human perception: although such models can predict human categorizations and brain activity with some sets of images, they appear to do so using features different from those humans use and they cannot reproduce key findings from psychological research[28,31]. It is likely that 3D generative models of the physical world are necessary to accurately predict and explain human perception. For example, human-engineered computer graphics models better explain visual perception and their parameters (features) can be easily controlled[31–34]. One trade-off, however, is that their features are restricted to the imagination of human engineers. In the future, intermediate layers of such generative models could be used as a feature space.

Using a word embedding in the analysis also entails caveats. As with the choice of a visual feature space, a semantic space that is not a good fit to how human representations are structured might hinder the recovery of accurate representations for some categories. For example, if the categories forest and tree are not close in the semantic space but forest and house happen to be, the representation of the concept forest might be biased toward the representation of the concept house" especially if the word forest itself has not been responded much. These word embeddings are trained on billions of words, and they excel at performing various types of natural language processing tasks[66,67], indicating that they capture real and useful semantic relationships between the words. Moreover, they can capture at least some aspects of semantic representations in the human brain[18,50,68–70]. Most importantly, our analyses suggest that the word embedding modeled the human representational space reasonably well: the representations we uncovered were similar to representations uncovered without using an embedding, and the embedding allowed us to recover representations of concepts even without using the concept's name in the analysis (Fig. 5). It is possible however that embeddings trained on different data (e.g., visual co-occurrences, visual similarities or human similarity judgments)[43,71–73] would model the structure of the representations even better. Although it would likely require more data, a nonlinear regression could also provide a better mapping between visual and semantic features. The method

may not be optimal. Although pixels can allow the recovery of any image, using more complex features reduces the space of possible images and introduces some biases. These apparent drawbacks are balanced by other, more positive consequences. First, the greater expressiveness of pixels is only theoretical: in practice, it would require an unreasonable number of trials to recover representations of complex targets. Second, we are not simply seeking to reconstruct existing 2D images. As noted above, representations of high-level concepts live in a more abstract feature space, and so we are first aiming to recover representations in that feature space, and only afterwards recreating an image from that representation. This last step could be modified to recreate different—or even multiple—images, without altering the reverse correlation aspect of the paradigm that operates in the abstract feature space. Finally, CNN features are likely more appropriate features to sample than pixels if our goal is to retrieve representations of natural concepts. In computer vision, CNNs are typically trained to categorize objects using millions of image-label pairs. In this process, the different layers of the networks learn increasingly abstract features that ultimately map combinations of raw pixels to one of multiple categories. These networks perform this task with near-human or superhuman accuracy[57–59], suggesting that the features they learn are a good representation of natural images. Moreover, CNN

could also be extended to phrases or sentences by asking participants to respond with these and by using a sentence embedding (e.g., refs. [74],[75]) in the analyses. Currently, our use of a word embedding limits us to concepts that can be defined with one-word labels. This extension might require other improvements to the paradigm, but if successful, would open new possibilities. Notably, we could investigate whether we can generate representations corresponding to novel combinations of objects and how these are related to the representations of the individual objects. Various compositions of individual concepts (e.g., "hat on a dog" from "hat", "on" and "dog") could be reconstructed, even if these simpler concepts have only been written in response to distinct stimuli.

A key advance of this work is the ability to investigate conceptual representations in their totality (modulo the caveats above), rather than focussing on a few specific concepts. Characterizing whole representational spaces is a central theme of cognitive computational neuroscience[76]. Unlike representational similarity analysis (RSA)[77], our method not only characterizes the internal structure of these spaces (i.e., the representational geometry), but also the exact positions of the representations within them (i.e., the representational content, or the feature values associated with each concept). Note that such positions cannot be derived from representational distances; there are in fact an infinity of representational spaces resulting in the same representational geometry. This additional information is necessary to reconstruct and predict representations. In that sense, our method is more akin to encoding models used with neural data[78],[79]. However, it is unclear whether encoding models fit exclusively on brain activity could retrieve unbiased mental representations of visual categories. Our work is also related to other studies assessing the contribution of mid-level features to the representations of high-level categories such as real-world size and animacy[80–82]. However, these studies did not attempt to relate specific features with categories, nor did they target a large number of categories. This important distinction is what enables us to analyze and reconstruct the contents of visual representations on a large scale across conceptual space.

In addition to investigating individual differences in representational content, this approach could be used to investigate representational differences across development, expertise levels, and cultures. It could also provide a platform for answering questions about mental representations more generally, such as differences between observed and optimal representations, the relationship between category and exemplar representations, and the influence of different experiences on represented content. The stimulus synthesis procedure alone could also be used in conjunction with other behavioral tasks or neuroimaging experiments to answer additional questions. The objective function could be modified to edit the features of otherwise natural images in a systematic way or to synthesize artificial images with varying representational similarity[83]. Additional analyses could also be performed on the data collected from this experiment. Notably, even though CNN features were manipulated, we analyzed other visual features of the reconstructions including pixels, color channels, and the Fourier power spectrum (see Fig. S1).

In summary, we mapped visual features to semantic features to characterize the representational space of natural categories. This allowed us to reconstruct images to visualize representations of many concepts in that space, even in individual participants. We also showed that conceptual representations in humans differ from those in the neural network used to synthesize the stimuli. Finally, the reconstructed conceptual representations generalized to new stimuli and to a new task. This new framework enables a global characterization of representational content.

## Methods

All studies comply with all relevant ethical regulations and were conducted under a protocol approved by the Yale University Institutional Review Board (IRB). Participants across all studies provided informed consent before participating in the study.

### Main experiment

The following experimental details were preregistered (https://aspredicted.org/QIN_RFK; on October 12th, 2020): sample size, exclusion criteria, experimental design, word embedding, CNN and layer, regression analyses, image reconstruction procedure, and analysis of network representations. We deviated from the preregistration in two minor ways for optimality reasons: we used a different word embedding and we used an external dataset (Visual Genome) instead of a cross-validation procedure to choose concepts to reconstruct. See supplemental information for originally planned analyses with the word2vec word embedding (Fig. S2). All code was written in Python, using the NumPy (https://numpy.org)[84] and PyTorch (https://pytorch.org) libraries.

**Participants.** Participants (healthy adults, aged 18–35, with normal or corrected-to-normal vision) were recruited via the Prolific platform[85] (https://prolific.co) until we got a final sample size of 100 after exclusions. No statistical method was used to predetermine sample size. Sex and gender of participants were not obtained and were not considered to be relevant to the study. Participants that did not complete the experiment from beginning to end or that wrote too many non-concrete words (more than 25% of words with a concreteness rating inferior to 4, as assessed by ref. [86]) were excluded. Participants provided informed consent to a protocol approved by the Yale IRB and were compensated 5$ for their participation.

**CNN-noise stimuli.** For stimulus synthesis, we used an instance of the ResNet-50 convolutional neural network[87] that was trained for object categorization on the ImageNet Large-Scale Visual Recognition Challenge (ILSVRC) 2012 dataset[58] while being robust to adversarial examples (L2 adversarial loss, eps = 3.0; available at https://git.io/robust-reps)[88]. Robust training seems to be necessary (along with the optimization procedure described below) to synthesize clear non-adversarial features[88],[89]. Specifically, we chose the 37th layer of the network (last layer of the 4th stage) as our feature space. We aimed to choose a layer that would represent relatively high-level features (and allow us to synthesize evocative stimuli) while not being too close to whole objects or concepts (which would constrain our search space too much).

The activations of the layer to all images of the ILSVRC 2012 validation set were first collected and averaged within channels (therefore ignoring spatial location). The resulting channel activations were standardized across images and the covariance matrix of the channels was estimated with optimal shrinkage toward the diagonal matrix[90]. A ZCA/Mahalanobis whitening transform was estimated and applied to the standardized activations to decorrelate the features. The distribution of activations to the validation set images was estimated for each feature in this whitened space using Gaussian kernel density. For each stimulus to synthesize, random values were drawn from the estimated distributions for all features; these random feature values were then colored (i.e., the inverse of the whitening transform was applied) and unstandardized to get back to the original space of the layer channels: these are our target CNN feature values.

We then synthesized a stimulus from these feature values using iterative optimization. We refer to this stimulus as a CNN-noise"-stimulus because it is associated with approximately random values in the space of CNN features (in contrast to pixel noise which would represent random pixel luminance variations). Specifically, we adapted the activation maximization algorithm from ref. [15]. That is, we first set random complex Fourier coefficients, drawing from a Gaussian distribution with a standard deviation of 0.01. Then, on each iteration of the optimization procedure, (i) the coefficients were normalized

according to their frequency (1/f scaling); (ii) they were inverse Fourier transformed; (iii) the color channels of the resulting image were decorrelated using the Cholesky decomposition of their covariance matrix estimated using the ILSVRC 2012 training set; (iv) the resulting image was fed to the CNN; (v) the mean squared error between the activations of the layer of interest and the target CNN feature values was computed; and (vi) the gradients with respect to the Fourier coefficients were estimated and applied using the Adam optimizer[91,92] (weight decay = 0.1; learning rate = 0.05; $\beta_1 = 0.9$; $\beta_2 = 0.999$). After 1500 iterations, the final Fourier coefficients were inverse Fourier transformed and the color channels of the resulting image were once again decorrelated: this results in the final optimized CNN-noise stimulus. The median reconstruction $R^2$ (comparing the target CNN feature values with the CNN feature values of the final images) was 0.93. The final CNN feature values rather than the original ones were used in all analyses. All stimuli used in the experiment (excluding the practice trials) were different, even across participants, to sample the feature space exhaustively.

**Experimental design**. The experiment was programmed using the PsychoPy (Python) and PsychoJS (Javascript) libraries (https://www.psychopy.org)[93], and it was carried out online on the Pavlovia website (https://pavlovia.org). Prior to the task, participants were provided with detailed written instructions. They were told that "dream-like pictures" would be shown to them, that "unclear or distorted objects, object parts and textures may be seen in the pictures" and that they would need to indicate the "objects or concrete things" that they saw, between 1 and 3 for each picture (to limit the duration of the experiment). They were further asked to be as concise as possible and to use only one noun per thing whenever possible. Participants were asked to stay at one arm's length from their screen (the experiment could not be performed on mobile devices), and they were asked to match the length of a segment on the screen to the length of a debit/credit card they owned to calibrate the size of the stimuli to 6 degrees of visual angle. Participants then needed to perform 5 practice trials (identical across participants) before completing 100 experimental trials (different across participants) in blocks of 20. Between blocks, participants were reminded of the instructions; they could start the next block whenever they were ready to do so. At the end of the experiment, a message was displayed, and participants were redirected to the Prolific website.

On each trial, a mid-gray screen was shown for 200 ms, followed by the CNN-noise stimulus centered on a mid-gray background for 5 s, followed by a mid-gray screen. It is at that moment that participants needed to indicate the visual concepts (min. 1) they perceived in the previous stimulus. Participants had no time limit to enter the labels but were told to answer quickly. To speed up the process, words were automatically suggested based on the characters entered and participants could press a key to accept the suggestions. The words that could be suggested initially consisted of the words with a concreteness rating of at least 4 in the database compiled by ref. 86 and they were suggested based on their frequency as compiled in that database. The suggestions were then adapted to the words written by the participant during the experiment. These automatic and adaptive word suggestions were implemented using the Predictionary JavaScript library (https://github.com/asterics/predictionary). Although participants' responses could have been biased by the automatic suggestions on some occasions, we think that this is unlikely to have had a significant impact on the results for two reasons. First, participants likely perceived the concept they intended to report while the stimulus was being shown, before they could begin responding; to accept a different suggestion would require both that it start with the same letters and that it be a better expression of the perceived concept. Second, autocomplete suggestions were not accepted often in practice, a median of only 9.3% of the time across participants.

**Visual-semantic matrix**. We first removed all stopwords (using the list from the NLTK Python library[46]; https://www.nltk.org), all one-character words and all numbers from the responses. Then, we used the SymSpell library (https://github.com/wolfgarbe/symspell) to automatically correct (using a maximum edit distance of 2) any word that was not recognized by the word embedding used in the analyses (see below). When there was more than one possible correction, we prioritized suggestions that were visual words (defined as being present in the list of labels of the Visual Genome natural image database[51]; https://visualgenome.org). Words that could not be corrected or that were still unrecognized after correction were removed. All remaining words were then transformed to semantic feature values using a pre-trained GloVe word embedding. This embedding maps a vocabulary of 400,000 words to 300-dimensional vectors based on their co-occurrence in large text corpora; related words typically have similar vectors and different dimensions encode different semantic aspects[70]. When a label was comprised of more than one word, these were considered as a single "word" in the rare case that they were recognized as such by the word embedding; when not recognized as a single word, the words were split and separately transformed into semantic feature values. For each stimulus (shown only once to one observer), the vectors of semantic feature values of all reported words were then averaged into one vector, to give equal weights to all trials regardless of the number of responses and to preserve the random sampling of visual features by not repeating the same visual features for multiple words in the creation of our visual-semantic matrix.

Rather than using only the sampled CNN layer for the analyses, we chose to use both the layer that was sampled and a slightly higher layer (43rd layer), following testing on pilot data. We averaged activations within each channel and concatenated channel activations from both layers: these were our new CNN feature values. Principal component analyses (PCA) with whitening were then applied to both the CNN feature values and the semantic feature values across trials. This was done both to decorrelate the features and to reduce the dimensionality of the data. In both cases, the minimum number of components to explain 90% of the variance was kept (CNN: 213, semantic: 127). Linear associations were then inferred by performing an outer product of the two trial × principal components (PC) matrices; this is equivalent to a multivariate multiple linear regression given that variables are random and uncorrelated[7,8,94]. The result was a CNN PC × semantic PC matrix indicating how much each CNN PC correlates to each semantic PC (Fig. 2b). We repeated these analyses 1000 times while randomly permuting the vectors of semantic feature values across trials (thus disrupting the potential associations between responses and stimuli) to establish a null distribution of matrices. To assess the statistical significance of the visual-semantic associations, we computed the maxima across coefficients of the null matrices and we extracted the percentile of the actual value in this distribution of maxima as the p value[95]. Such a statistical test allows us to test the statistical significance of all coefficients while relying on fewer assumptions than parametric tests and correct for multiple comparisons using the optimal maximum statistic method. To visualize semantic PCs, we selected the words, among all words responded in the experiment, that loaded most strongly for that PC (Fig. 2c). To visualize CNN PCs, we maximized the values in the direction of the vector of feature values corresponding to the CNN PC (using the caricature objective function, see Methods: Reconstruction of mental representations; Fig. 2c). The optimization was carried out according to the procedure described in ref. 15. Additionally, we selected the words that had the highest cosine similarity between their semantic feature values and the semantic feature values associated with the CNN PC (Fig. 2d).

**Reconstruction of mental representations**. To visualize a mental representation of a concept, we first computed the CNN feature values associated with this concept. To do so, we obtained the concept's

semantic feature values using the word embedding, transformed the semantic feature values to semantic PC values using the fitted semantic PCA, transformed the semantic PC values to visual PC values using the visual-semantic matrix, and transformed the visual PCs to layer channels using the fitted visual PCA. Note that these CNN feature values can be outside the distribution of values in natural images.

We then iteratively optimized an image from these target CNN feature values using a procedure similar to the one employed to generate stimuli, except for two important things. First, we maximized the caricature objective function[96] (Eq. (1); Cammarata, Olah & Satyanarayan, in preparation) instead of minimizing the mean squared error:

$$\mathbf{p}^* = \arg\max_{\mathbf{p}} \langle \mathbf{y}, \phi(\mathbf{p}) \rangle \left( \frac{\langle \mathbf{y}, \phi(\mathbf{p}) \rangle}{||\mathbf{y}|| ||\phi(\mathbf{p})||} \right)^{\alpha} \quad (1)$$

where $\mathbf{y}$ is the target vector of feature values, $\mathbf{p}$ are the pixel values of an image, $\phi(\mathbf{p})$ is the feature vector associated with these pixel values, $\mathbf{p}^*$ are the pixel values of the reconstructed image and $\alpha$ is a free parameter. To summarize, this objective function consists of the dot product between the target CNN feature values and the CNN feature values of the synthesized image, multiplied by the cosine similarity between these vectors of feature values weighted by some arbitrary exponent $\alpha$. This has the effect of maximizing feature values in the direction of the target CNN feature values in high-dimensional feature space, instead of trying to match it exactly; the cosine similarity term ensures that the direction stays reasonably close to the target direction (the strength of this constraint can be varied by adjusting the $\alpha$ exponent; here, $\alpha$ was set to 4, following testing on pilot data). We used this objective function because the magnitude of the target feature vector is arbitrary, the optimal magnitude for reconstruction is unknown, and maximizing feature values allows us to visualize the important features of the representation most clearly. Images were optimized for 2000 iterations using a learning rate of 0.05. A second important difference is that we use transformation robustness in this reconstruction procedure; that is, at each iteration of the optimization process, the image is subjected to small rotations, translations and homothecies prior to be fed to the network. This has been found to reduce high-frequency artifacts and improve the clarity of the optimized images[15]. Because of these changes, there may be small baseline visual differences between reconstructions and stimuli.

To visualize the uncertainty intrinsic to the representations, we synthesized images from the CNN feature values that correspond to the lower and upper bounds of 95% confidence intervals around the CNN feature values associated with some concepts (Fig. 5c). To do so, we inferred new visual-semantic matrices while randomly resampling trials (in the same way for both semantic and CNN feature values) 1000 times with replacement. We then computed the CNN feature values associated with a given concept using each matrix. For each CNN feature, the lower and upper bounds of the confidence interval corresponded to the 2.5th and 97.5th percentiles of this bootstrap distribution. Images were then synthesized from these CNN feature values. Null reconstructions were also synthesized in a similar way, by using the CNN feature values associated with a given concept according to a null visual-semantic matrix obtained by permuting the trials (see Methods: Visual-semantic matrix).

**Investigation of the effect of the semantic embedding.** To assess the effect of responses containing the concept's name, we repeated for each target concept the creation of a visual-semantic matrix as in the main analysis but ignoring responses with the concept's name. We then computed the concept's CNN feature values using this matrix and reconstructed representations from these using the same procedure as was used in the main analysis (Fig. 5d). Finally, we correlated the feature values with the CNN feature values associated with

the concept according to the main analysis (Fig. S3a). To assess the statistical significance of the correlations, we computed them again but using the null visual-semantic matrices (obtained while randomly permuting the trials 1000 times). The null correlations obtained using this randomization procedure were then used to z-score both observed and null correlations. Finally, we computed the maxima across concepts of the null values and we extracted the percentile of the actual value in this distribution of maxima as the p value[95]. We also computed the bootstrapped estimates of the standard errors of the correlations by repeating the computation of the correlations 1000 times with the dimensions of both variables resampled with replacement.

To assess the impact of the semantic embedding, we repeated the main analysis but replaced the semantic embedding with a binary embedding representing whether each word (among the 369 words named at least 10 times) was answered or not on a given trial. This allowed us to directly obtain the CNN feature values associated with each of these words. We then correlated these feature values with the CNN feature values of the concepts according to the main analysis (Fig. S3b) and we reconstructed the representations of the 10 most-named concepts using the same procedure as was used in the main analysis (Fig. 5e). To assess the statistical significance and compute the standard errors, we used similar procedures as for the previous analysis.

**Prediction of semantic content.** To predict the semantic content of new images, we obtained the CNN feature values associated with these images by feeding them to the CNN, transformed these into visual PC values using the fitted visual PCA, transformed the visual PC values to semantic PC values using the visual-semantic matrix, and transformed the semantic PC values to semantic feature values using the fitted semantic PCA. We then compared these semantic feature values to the actual semantic feature values from the responses associated with the image using cosine similarity (Fig. S4a). To assess statistical significance of the mean cosine similarity, we repeated the above procedure 1000 times while using as responses labels randomly drawn (with replacement) from all labels associated to the tested images, and we extracted the percentile of the actual value in this null distribution as the p value. We also computed the bootstrapped estimates of the standard errors by repeating the computation of the cosine similarities 1000 times with the dimensions of both vectors resampled with replacement.

**Prediction of stimuli.** To predict the stimuli associated with a specific concept, we used the visual-semantic matrix from the main experiment and the set of stimuli and responses from the Individual Representations experiment. For each of the 10 most-named concepts (grass, sky, tree, dog, bird, water, animal, snake, building, eyes), we created a Boolean vector of 0 s and 1 s indicating which stimuli were associated with the concept at least once. Then, we created another Boolean vector representing our predictions of which stimuli were associated with the concept solely based on its visual features. To do so, we computed how much the CNN feature values of each stimulus correlated to the CNN feature values associated with the concept (as obtained with our visual-semantic matrix) and thresholded this vector of correlation coefficients so that the number of stimuli predicted as containing the concept matched the number of stimuli containing it. We then quantified the degree of overlap between the two Boolean vectors by computing their Dice coefficient (Fig. S4b). To assess statistical significance, we computed null Dice coefficients by repeating this analysis 1000 times while randomly permuting the order of the stimuli in one variable. We then z-scored the observed and null Dice coefficients using the mean and standard deviation of the null Dice coefficients, computed the maxima across concepts of the z-scored null Dice coefficients, and extracted the

percentile of the actual value in this distribution of maxima as the *p* value[95].

**Prediction of behavioral similarity judgments.** We used the openly available dataset (https://osf.io/um3qg/) used in ref. 50. In that study, participants are asked to place words describing visual concepts on a 2D plane according to their semantic similarity (total of 60 words). Subsets of words are shown on each trial in an adaptive fashion, and a representational dissimilarity matrix (RDM) is derived from these judgments using the inverse multidimensional scaling (MDS) algorithm[97]. We rank-transformed RDMs from all participants and averaged them: this is the behavioral judgments RDM. To derive the visual RDM, we obtained the CNN feature values associated with each of the 60 concepts using the word embedding and our visual-semantic matrix (see Methods: Reconstruction of mental representations) and we computed the correlation distances (1 – Pearson correlation) between all vectors of feature values. We used a similar procedure to derive the semantic RDM, using the semantic feature values associated with each concept instead of the CNN feature values. To assess how well the visual RDM explained the behavioral RDM (and how well the semantic RDM explained the behavioral RDM), we computed the Spearman correlation between the vectorized upper triangles of the matrices. We used Spearman correlation because only the ranks of the distances can be meaningfully compared between matrices[77,98]. We performed a randomization test to assess statistical significance: the Spearman correlation was computed 1000 times while randomly shuffling rows and columns of the behavioral RDM[77] and we extracted the percentile of the actual value in this null distribution as the p value. We used a nonparametric test because it relies on fewer assumptions than parametric tests and allowed us to keep a coherent statistical approach across analyses. To assess whether the visual RDM correlates more to the behavioral judgments RDM than the semantic RDM, the Spearman correlation was computed for the semantic RDM and the difference between both correlations was computed; a randomization procedure similar to the one described above was used with the correlation difference to test its statistical significance. We also computed the bootstrapped estimates of the correlations by re-calculating the correlations 1000 times after randomly resampling the rows and columns of the matrices with replacement (and eliminating the off-diagonal zero entries).

**Investigation of the representations of the deep neural network.** We had the deep neural network perform the same experiment with the same stimuli as the participants in the main experiment. (Note that in this section, "deep neural network" denotes the neural network under study, while "CNN feature values" denotes the vectors of visual features that constitute the mental representations. Since both the representations of humans and the network are studied, there are CNN feature values associated with both humans and the DNN.) The labels associated with the top 3 ImageNet classes predicted by the network (i.e., the 3 classes with the highest probability) were taken as its responses. When a label comprised multiple synonyms, we used the first one, which was usually the simplest and most common. When a response was not recognized by the word embedding, we went up one level in the WordNet hierarchy to which ImageNet classes are mapped (e.g., from otterhound to hound) and tried again. We used this procedure because ImageNet classes (and therefore the DNN's responses) are sometimes too specific for the word embedding's vocabulary. We analyzed the data in the same way as with the participant data (number of semantic PCs = 118) and we reconstructed the representations of the 10 concepts most common in the Visual Genome database. To visualize the representational spaces, the CNN feature values associated with the 100 concepts most common in the Visual Genome database were projected on a two-dimensional plane using t-SNE[99] with correlation distance as metric

(perplexity = 15). The 2D plot of the DNN's representations was further aligned as best as possible with the plot of human representations using Procrustes transformations.

To investigate differences between the groups (DNN vs. humans), we divided the data in halves and computed the visual-semantic matrices for each half and group. We then projected the matrices to an independent semantic space: to do so, we computed for each half and group the CNN feature values associated to each one of the 100 most common concepts from the Visual Genome database (each concept can be viewed as a dimension of a new semantic space). The set of these CNN feature values summarizes, once flattened, the visual representations of either DNN or humans. We computed the Pearson correlation between the human visual representations in half 1 and the DNN visual representations in half 2, and the correlation between the human representations in half 2 and the DNN representations in half 1. We then averaged these two correlation coefficients to obtain a between-groups correlation value. We also computed a within-groups correlation: this is the average of the correlation between the human representations in half 1 and the human representations in half 2, and the correlation between the DNN representations in half 1 and the DNN representations in half 2. The difference between these two average correlations reflects the degree to which the two sets of representations (DNN and human) are unique[100]. To assess statistical significance, these analyses were repeated but using the null visual-semantic matrices (obtained while randomly permuting the trials 1000 times) and we extracted the percentile of the actual value in this null distribution as the p value. To quantify the uncertainty around these correlations, we repeated the analyses by inferring new visual-semantic matrices after randomly resampling trials 1000 times with replacement (in the same way for both semantic and CNN feature values).

## Validation experiment #1: categorization of reconstructions

The goal of this experiment was to validate the reconstructions of mental representations in separate participants. The experiment was preregistered at https://aspredicted.org/SLR_KFB on June 24th, 2021. There were no deviations to the preregistered protocol.

**Participants.** Fifty new participants (healthy adults, aged 18–35, with normal or corrected-to-normal vision) were recruited via the Prolific platform[85]. No statistical method was used to predetermine sample size. Sex and gender of participants were not obtained and were not considered to be relevant to the study. Participants provided informed consent to a protocol approved by the Yale IRB and were compensated 3.25\$ for their participation. Two participants were excluded and replaced because of mean accuracies more than 3 standard deviations away from the group mean.

**Stimuli.** Reconstructions validated include the reconstructions for the 250 concepts that were most named during the main experiment along with the 100 most frequent concepts in the Visual Genome database[48] that were named less than 10 times in the main experiment. Reconstructions were obtained using the procedure described above (see Methods: Reconstruction of mental representations).

**Experimental design.** The experiment was carried out online on the Pavlovia website (https://pavlovia.org). We used a similar validation procedure as was used in ref. 24. Prior to the task, participants were provided with detailed instructions: they were told that each depicted concept may be unclear, may occupy the whole image or only a part of it, and may be depicted many times across the image. Participants were also told to keep in mind the different potential meanings of each label. On each trial, a mid-gray screen was shown for 200 ms, followed by a randomly selected reconstruction centered on a mid-gray background. After 1 s, two labels appeared at the top of the screen: the true

label and one wrong label chosen randomly among the labels of all other concepts validated in the experiment (different across participants). The two labels were randomly assigned to either the top left corner or the top right corner. Participants needed to indicate at that moment which label most likely corresponded to the image; they could answer with two keyboard keys. There was no time limit, but participants were told to answer quickly. The 350 reconstructions were shown in 7 blocks of 50; participants could take short breaks between blocks. The presentation order of the stimuli was randomized across participants.

**Data analysis**. We computed the mean accuracies across participants for each concept. To assess their statistical significance, we computed them again while randomly resampling the participants with replacement 50,000 times and assessed whether at least 99.993% (one-tailed 95% significance threshold with Šidák correction for multiple comparisons across the 350 concepts) of that distribution was above the chance level of 50%. For concepts for which the estimated accuracy separating the rightmost 99.993% of the distribution from the rest was between 40% and 60%, the procedure was repeated with 500,000 iterations to improve the precision of the estimation. This statistical procedure estimates the effect for the population of participants while correcting for multiple comparisons across concepts[101]. To assess the statistical significance of the mean accuracies for all concepts or subsets of concepts, we permuted the trials randomly 1000 times to establish null distributions. We quantified the p value as the percentile of the observed mean accuracy in the null distribution. We also assessed the statistical significance using a variant of the above bootstrap procedure (but averaging across multiple objects and using a 95% threshold). Both statistical methods gave similar results (all mean accuracies significant with $p < 0.001$, one-tailed).

## Validation experiment #2: Labeling of reconstructions
The goal of this additional experiment was to further validate the reconstructions. This experiment was not preregistered.

**Participants**. Fifty new participants (healthy adults, aged 18–35, with normal or corrected-to-normal vision) were recruited via the Prolific platform[85]. No statistical method was used to predetermine sample size. Sex and gender of participants were not obtained and were not considered to be relevant to the study. Participants provided informed consent to a protocol approved by the Yale IRB and were compensated 7.34$ for their participation.

**Stimuli**. The images validated were the reconstructions for the 100 concepts that were most named during the main experiment. Reconstructions were obtained using the procedure described above (see Methods: Reconstruction of mental representations).

**Experimental design**. The experiment was carried out online on the Pavlovia website (https://pavlovia.org). Prior to the task, participants were given detailed instructions: They were told that each depicted concept may be unclear, may occupy the whole image or only a part of it, and may be depicted many times across the image. Participants were also told that each image is associated to a different label, and that synonyms and different spellings all count as different words. On each trial, a mid-gray screen was shown for 200 ms, followed by a randomly selected reconstruction centered on a mid-gray background for 2 s, followed by another blank screen on which participants could write three guesses about the label of the reconstruction. There was no time limit, but participants were told to answer as fast as possible. When they were done entering their third guess, the next reconstruction was shown. The 100 reconstructions were shown in 5 blocks of 20; participants could take short breaks between blocks. The presentation order of the stimuli was randomized across participants.

**Data analysis**. We first corrected spelling mistakes in the participants' responses using the same procedure that was used for the main experiment. We then analyzed how many concepts were labeled correctly. We considered plural and singular forms of a word to be equivalent. For each reconstruction, we verified whether the correct label was the one provided most commonly by participants. This resulted in a vector of binary values (one for each concept). We summed this vector to obtain the number of successful concepts. To obtain a confidence interval around this number, we computed it again after randomly resampling the binary vectors 1000 times with replacement. To assess whether the number of successful concepts obtained was significant, we repeated the above analysis 1000 times after randomly permuting participant responses across trials (reconstructions). This resulted in a null distribution of 1000 numbers of correct concepts. We extracted the percentile of the actual value in this null distribution as the p value. We then analyzed how many concepts were generally well labeled with semantically close responses, even if these were not the concept's exact true label. To do so, for each concept, we fit a first-degree polynomial equation between the semantic distance of each unique response to the true label and its frequency. We then transformed the slope coefficient into a t value and computed its significance using the Student's t distribution, Bonferroni-correcting for the 100 statistical tests.

## Validation experiment #3: Labeling of real and null reconstructions
The goal of this additional experiment was to further validate the reconstructions and compare them to null reconstructions. This experiment was not preregistered.

**Participants**. Twenty-five new participants (healthy adults, aged 18–35, with normal or corrected-to-normal vision) were recruited via the Prolific platform[85]. No statistical method was used to predetermine sample size. Sex and gender of participants were not obtained and were not considered to be relevant to the study. Participants provided informed consent to a protocol approved by the Yale IRB and were compensated 6.00$ for their participation. Six participants were excluded and replaced because they provided responses that were more than one word for 90% of trials or more.

**Stimuli**. The images validated were one real reconstruction and three randomly selected null reconstructions (reconstructions obtained with null visual-semantic mappings) for a random subset of 45 concepts among the 100 concepts that were most named during the main experiment (total of 180 images). Reconstructions were obtained using the procedure described above (see Methods: Reconstruction of mental representations).

**Experimental design**. The experiment was carried out online on the Pavlovia website (https://pavlovia.org). Prior to the task, participants were given detailed instructions: They were told that each depicted concept may be unclear, may occupy the whole image or only a part of it, and may be depicted many times across the image. On each trial, a mid-gray screen was shown for 200 ms, followed by a randomly selected reconstruction centered on a mid-gray background for 2 s, followed by another blank screen on which participants could label the reconstruction. There was no time limit, but participants were told to answer as fast as possible and to write only one word per trial. The 180 reconstructions were shown in 5 blocks of 36; participants could take short breaks between blocks. The presentation order of the stimuli was randomized across participants.

**Data analysis**. We corrected spelling mistakes in the participants' responses using the same procedure as the main experiment and

considered plural and singular forms of a word to be equivalent. We computed, for each concept, whether: (1) the correct label was provided more often for the real reconstruction than for null reconstructions on average; (2) the semantic distance of responses to the correct label was lower for the real reconstruction than for null reconstructions on average; (3) the entropy of the response probability distribution was lower for the real reconstruction than for null reconstructions on average; (4) the trace of the covariance matrix of the semantic features of responses was smaller for the real reconstruction than for null reconstructions on average. The resulting vectors for these four metrics (one binary value per concept) were then summed to obtain the numbers of successful concepts reported in the main text. To get uncertainty estimates around these numbers, we recomputed the metrics after randomly resampling the binary vectors 1000 times with replacement. Confidence intervals were obtained by extracting the 2.5th and 97.5th percentiles of this bootstrap distribution. To test the statistical significance of the numbers of successful concepts, we calculated a null distribution for each metric by randomly permuting the 45 concepts (or in the case of analyses 3 and 4, randomly permuting all 4500 responses: 25 participants × 45 concepts × 4 images) 1000 times. We extracted the percentile of the actual value in this null distribution as the $p$ value.

### Individual representations experiment

The goal of this additional experiment was to reconstruct the mental representations of individual observers. This experiment was not preregistered.

**Participants.** We recruited eight new participants (healthy adults, aged 18–35, 5 women and 3 men, with normal or corrected-to-normal vision) from the Yale University community. No statistical method was used to predetermine sample size. Sex and gender of participants were not considered in the study design or analyses because this was not relevant to the questions under investigation. Participants provided informed consent to a protocol approved by the Yale IRB and they obtained a 45$ Amazon gift card for their participation. No participant was excluded prior to analyses.

**Stimuli.** Stimuli were created using the same procedure as in the main experiment. We created 750 stimuli (in addition to 5 stimuli for practice trials) that were used for all participants. Stimuli were identical across participants to facilitate comparisons between them.

**Experimental design.** The experiment was carried out online on the Pavlovia website (https://pavlovia.org). The experimental design was the same as in the main experiment except that each participant performed 6 sessions of the experiment, that there were 125 stimuli per session (shown in 5 blocks of 25), and that participants were not redirected to any website at the end of the experiment. The presentation order of the stimuli was randomized across participants.

**Data analysis.** For each participant, we created a visual-semantic matrix as described above (see Main experiment: Visual-semantic matrix; number of CNN PCs = 150; number of semantic PCs varied between 55 and 120). We reconstructed individual visual representations using these matrices (Fig. 7; see Main experiment: Reconstruction of mental representations). To visualize similarities and dissimilarities between these representations, they were projected on a two-dimensional plane using t-SNE[96] with correlation distance as metric (perplexity = 15).

To assess inter-individual differences in the representations, we used a similar procedure as for the investigation of the network's representations, but we projected the visual-semantic matrices to the independent semantic space defined in the main experiment, and we

computed the uniqueness coefficients for each pair of participants and averaged them[100].

### Reporting summary

Further information on research design is available in the Nature Portfolio Reporting Summary linked to this article.

## Data availability

All raw and preprocessed data generated and analyzed during this study are available at https://osf.io/mp3s6/[102]. The following publicly available data were also used in the study: Behavioral dataset on semantic word arrangement (https://osf.io/um3qg/)[50]; Visual Genome dataset (https://homes.cs.washington.edu/~ranjay/visualgenome/index.html)[48]; GloVe word embedding (https://nlp.stanford.edu/projects/glove/)[67]; pretrained adversarially robust ResNet-50 (https://github.com/MadryLab/robust_representations)[89]; and ImageNet Large-Scale Visual Recognition Challenge (ILSVRC) 2012 dataset (https://www.image-net.org/challenges/LSVRC/2012/)[58].

## Code availability

All code is available at https://github.com/laurentcaplette/Representation-reconstruction[103].

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

## Acknowledgements

We thank Jeffrey Wammes for help with the initial image synthesis efforts and Frédéric Gosselin for helpful discussions about various aspects of the paradigm. We are grateful for funding from the National Science Foundation (CCF 1839308; N. B. T.-B.) and Fonds de Recherche du Québec – Nature et Technologies (Postdoctoral Research Scholarship; L. C.). These funders had no role in study design, data collection and analysis, decision to publish or preparation of the manuscript.

## Author contributions

L. C. and N. B. T.-B. conceptualized and designed the experiment. N. B. T.-B. acquired funding. L. C. prepared the experimental code and stimuli. L. C. collected and analyzed data. N. B. T.-B. advised on data analyses. L. C. wrote the first draft of the manuscript. L. C. and N. B. T.-B. edited the manuscript.

## Competing interests

The authors declare no competing interests.
