## [Peer Review File · Nature Communications]

Computational reconstruction of mental representations using human behaviorEditorial Note: This manuscript has been previously reviewed at another journal that is not operating a transparent peer review scheme. This document only contains reviewer comments and rebuttal letters for versions considered at *Nature Communications*.

REVIEWER COMMENTS

Reviewer #4 (Remarks to the Author):

At the core of this manuscript is an interesting idea – use features generated from a deep neural network to probe internal mental representations in a similar way to prior reverse correlation studies using pixelated noise. While this is an interesting idea, and the instantiation of the method is potentially useful, overall I'm not convinced the study provides any compelling new insight into internal representations. There are clear limitations of the method and to some extent the authors are over-selling what can be concluded from the results.

I was brought in as a new reviewer and have seen some prior reviews and responses. The authors were certainly responsive to the prior comments and provided additional data and analyses. But I think there remain key issues that the authors need to address and the results, interpretation and limitations could be presented in a much more balanced way.

Specific Comments:

1) In prior studies using reverse correlation with pixelated noise, participants are typically making an explicit forced-choice categorization of noisy stimuli (e.g. "is there a letter 's' present?", "which facial expression is depicted?"). The idea is to determine which patterns of noise push the observers to make a particular categorization while the observers are holding in mind specific types of representation. In the current study, there is no forced choice categorization, the observers are not really holding any type of representation in mind and are just asked to come up with categories that might be associated with the DNN feature stimuli. People can obviously perform such a task, but just because observers can make an association does not mean it's tapping in to their internal representations, certainly not in the same way as the reverse correlation studies. For example, Zhou and Firestone (2019) showed that human observers can reliably anticipate the errors of DNN's on adversarial examples – people can do sophisticated processing and apply intuition to complex images. But it's not clear what results like this tell us about the nature of our internal representations.

2) There are clear benefits and limitations of the current approach over, say, the reverse correlation studies. Some of these issues were brought up in the prior reviews and the authors have made some revisions. But I don't think the authors are providing a balanced discussion of these issues – the authors really need to be moderating their claims. On top of some of the limitations that have already been raised, I'm wondering about the impact of the training task and stimuli for the DNN. The DNN was trained on an object categorization task with the Imagenet stimuli – these stimuli are primarily close up views of objects with minimal background. Interestingly, the responses of observers in the current study to the DNN feature stimuli often contain more scene features (e.g. grass, sky). Would comparable results be obtained if the authors used a DNN trained on scene categorization? One feature of such trained networks is they often have more object-like features in earlier layers of the network. What impact would this have on the estimated internal representations? Further, one aspect of Imagenet that is well recognized is that there are a disproportionate number of some categories. Such categories include dogs

and fish, which both feature prominently in the labels that observers gave for the DNN feature stimuli in the current study. Is it possible that the prominence of these labels in part reflects the nature of the Imagenet stimuli? The bottom line is that the choice of network, training task, and training stimuli are all factors that may affect the nature of the results reported here. Without explicitly testing their impact, it's hard to know how much the current results are influenced by them.

3) The DNN feature stimuli look like texturized stimuli akin to things produced by the Simoncelli texture synthesis algorithm and used by Bria Long and Talia Konkle to investigate the contribution of mid-level features to the organization of human ventral visual cortex (Long et al, 2018, PNAS; Long et al, 2016, JEP:Gen; see also similar work by Jagadeesh and Gardner, 2022, PNAS). It's not clear to me that the current work provides that much more insight than any of these studies. Mid-level features are associated with high-level visual categories and if you ask observers which categories might be associated with stimuli showing different types of features they can tell you.

4) Abstract: "there is currently no general framework for providing direct access to representations of high-level visual concepts". True, but this manuscript does not achieve this. And what would be "direct access"?

5) Throughout the manuscript, the authors tend to provide only qualitative overviews and summary statistics for the data. In many cases it's hard to understand exactly how the statistics were computed. I appreciate that the data will be made fully available, but I still think that more information should be provided in the manuscript or supplementary material. For example, the authors report that, "After corrections and removal of 152 invalid responses, this corresponded to 2,578 unique words, of which 369 were provided at least 10 times each (Figure 2a). The most frequent words were "grass" (607 responses), "sky" (592), "tree" (450), "dog" (355) and "bird" (355)". It would be helpful to see a frequency distribution and a list of all 369 words. Similarly, on Lines 199-207 it would be helpful to see more data – how many different words per reconstruction? How were the stats computed. These are just a couple of examples, but I think a fuller presentation of the data and greater clarity over the statistical testing throughout would strengthen the manuscript.

6) Participants were asked to indicate "objects or concrete things" that they could see in the DNN feature stimuli. Why was it limited to three? On lines 134-135, it is stated that "we asked observers to indicate all categories they perceived in the stimuli". This is inconsistent with the instructions described in the methods both in terms of the terms used and the fact that responses were apparently limited to three.

7) The initial validation with a 2AFC is a weak test – I suggest that much less emphasis be placed on this experiment.

8) Line 152: what's an invalid response?

9) Lines 322-324: "We reconstructed the network's representations in the same way as we did for the human representations. Resulting reconstructions often look superficially similar to reconstructions of human representations but with the concept rendered less clear or even unidentifiable (Figure 7a)." Why not run the same analyses as run for humans?

Overall, I think the manuscript presents some interesting ideas and an interesting approach, but I think the results and interpretation are very preliminary. Personally, I think the manuscript would be much stronger with a more balanced presentation of the limitations of the approach and more caution in the claims that are made.

Reviewer #5 (Remarks to the Author):

This paper provides a new approach for visualizing the mental representation of concepts, which is based on a reverse-correlation style approach that takes advantage of representations at a deep layer of a CNN. This seems like a novel approach that can be used in future work. The method and motivation for it is clearly laid out, and the authors provide a very comprehensive set of behavioral analyses. I appreciated the thorough control analyses that showed how the method generalizes across observers and stimuli. It looks like the comments that were left by the previous reviewers R1 and R2 have been addressed well.

I did have a few concerns and questions, which can hopefully be addressed in a minor revision. Here is a list of key points:

It would be helpful to clarify the scope of what is included as a “concept”. For example, most of the ones mentioned seem to be very concrete objects or “stuff”, but then there are colors like “white”? The concept “strange” is mentioned on line 424. Is “strange” or other adjectives like this included as a concept? What is the proportion of objects/stuff/adjectives within the set of concepts that you included? For objects, are the labels mostly basic-level (dog), or do they also include superordinate (animal) or subordinate (dog breed)?

Related to the above - in the 2AFC task where people are choosing between two labels for the reconstruction, is there ever a comparison between different types of things (like an object vs a color, or a superordinate category versus a basic-level category)? I can imagine this creating ambiguity, like if you have “animal” vs. “dog” for a dog image. How do you address this possibility?

For the result described on lines 208 – 214:

“Among the large number of concepts that could be reconstructed, it was still possible to distinguish between closely related and visually similar concepts ... singular and plural forms of the same concept could also be differentiated”

Please clarify what is meant by “differentiated” here. Is this just based on visual inspection of the images? Or is there an experimental validation/quantification of this claim? This observation is also repeated in the discussion lines 366 – 369. If it is just based on visual inspection, this claim is misleading and should be modified.

It would be worth discussing the possible limitations related to the dataset that the model is trained on (ImageNet). For example, it looks like the category “dog” shows up quite a lot in the responses (Figure 2 C-D), more so than other animals. This might be related to the fact that there are a lot of dog breeds in ImageNet. Does the set of concepts that you can reconstruct depend on what categories are present during model training?

In the introduction, line 101-102:

“An additional drawback of the reverse correlation and brain-based reconstruction approaches is that they focus on small sets of experimenter-defined stimuli.”

This doesn't seem true – some of the brain-based approaches that you cited do model a continuous stimulus space, so they can reconstruct stimuli that are not in the training set, not just a fixed set of stimuli. Same for the “superstimulus” approaches (Bashivan 2019, etc). Please be more precise with this statement and what is the novelty here.

For the first experiment, why did you average the feature vectors for all reported words together (lines 644 – 645) instead of keeping them separate? I am wondering if this would obscure any information, especially if the reported words for an image are very different. It looks like in some cases the random images have multiple distinct “things” in them, in different regions of the image, as opposed to just one concept. When people reported multiple concepts, were they usually very similar, or were they different? What if different people reported very different concepts? Please provide some more justification for this point.

How does the distribution of the “target” CNN activation vectors (line 678) differ from the distribution of the “random” CNN vectors used for the CNN-noise stimuli? I see that for the “random” CNN feature vectors, they are forced to have the distribution/covariance as the activations for real images. Do the “target” feature vectors for the reconstructions also have this property, or are they allowed to be less natural? From a glance at Figure 2, it looks like maybe the reconstructions are less natural in appearance than the CNN-noise images (maybe more high-freq artifacts?). Are there any consistent visual differences across these stimulus types?

I found the paragraph from lines 183 – 198 a bit confusing in how the different sets of images are presented. For example you have: “the 250 words most-named in the experiment and the top 100 most frequent words from the Visual Genome database labeled less than 10 times by participants.” But then, some of the 250 words in the first group are also part of visual genome? I got confused when you describe the “209” concepts, and then later the “47” concepts that are rarely or never named. Please clarify these different groups in the text.

For the second “validation” task described on lines 199 – 207, it seems like the accuracy is relatively low, even though it is above chance. I wonder if there is another way to assess accuracy here, as opposed to looking for an exact match between the written and actual concept, which is quite hard. For example could you compute how aligned are the embeddings of the written and actual concepts? This would capture if the guessed concept is “close” to the real concept.

Clarification on line 755: “We binarized this vector of correlation coefficients so that the number of stimuli predicted as containing the concept matched the number of stimuli containing it. “

I couldn't figure out what this means exactly, did you threshold the correlation coefficients to make them 0/1? Why was this done? Please clarify this method.

Figure 3 C is referenced in the text before A and B, these should be referenced in order.

Reviewer #6 (Remarks to the Author):

In this study, the authors create and validate an innovative update to the reverse correlation paradigm, which reconstructs mental representations for specific items at the level of pixels. This new method instead uses features from a visual CNN, and utilizes a category-agnostic approach where participants name multiple objects they see in CNN "noise" images. These object names are converted to semantic vector representations and linked to the CNN features used to generate the images so that novel images can be created based on the semantic vectorization of other words.

The authors perform a series of innovative analyses (e.g., extrapolating to unnamed object categories), show clear intuitive examples illustrating the effectiveness of their methods, and demonstrated the generalizability of their method. The paper is also well-written and relatively easy to parse.

Overall, I feel incredibly favorably about the paper. It is both very methodologically rigorous and theoretically innovative. It is clear the authors thought deeply, and each time I thought of a potential concern, they addressed it. I was also able to see their most recent set of responses to reviews at another journal, and I felt the authors addressed those comments well.

As a result, I have very little to add to the paper. I only have one very minor comment:
- "CNN feature values were more correlated within group" -- it would be helpful to have more clarification here. Is the comparison being shown here CNN within group vs. CNN-human across group, or is it CNN within group vs. human within group corrected by the across-group measure? It would be nice to see the comparison across all 3 measures (within CNN, within human, across CNN-human), maybe even as a small 3x3 correlation matrix in the figure.

REVIEWER #4

4.1 In prior studies using reverse correlation with pixelated noise, participants are typically making an explicit forced-choice categorization of noisy stimuli (e.g. “is there a letter ‘s’ present?”, “which facial expression is depicted?”). The idea is to determine which patterns of noise push the observers to make a particular categorization while the observers are holding in mind specific types of representation. In the current study, there is no forced choice categorization, the observers are not really holding any type of representation in mind and are just asked to come up with categories that might be associated with the DNN feature stimuli. People can obviously perform such a task, but just because observers can make an association does not mean it’s tapping in to their internal representations, certainly not in the same way as the reverse correlation studies. For example, Zhou and Firestone (2019) showed that human observers can reliably anticipate the errors of DNN’s on adversarial examples – people can do sophisticated processing and apply intuition to complex images. But it’s not clear what results like this tell us about the nature of our internal representations.

We agree that the task that participants had to perform in our study was different from most reverse correlation studies. Participants did not have to hold in their working memory a representation of a specific image. Explicitly holding a representation in mind would have been trickier to achieve with non-pictorial representations such as the ones we were targeting. However, all participants had abstract representations of multiple concepts encoded in their mind, even if they did not have to explicitly hold any specific one in their working memory. We believe that, for a participant to give a specific concept label to a stimulus, they must have perceived in it features that corresponded (at least partially) to their representation of the concept. Therefore, the features that we retrieve in our analysis are part of this representation. We agree with the reviewer that these visual features that participants associate with a certain concept might not be exactly the same as their internal representation of the concept. We have now nuanced our claims and discussed these limitations (see lines 452-461):

We must note however that our method is distinct from reverse correlation in some ways. Most importantly, participants do not have to hold a specific representation in their working memory while viewing the stimuli, before deciding whether the stimulus corresponds to their representation or not. This would be more difficult for participants to achieve with complex non-pictorial representations such as the ones we are targeting. Rather, we ask that they label each stimulus according to what they perceive most prominently in them. We assume that participants have abstract representations of multiple concepts encoded in their mind and that, to label the stimulus with a certain concept’s label, they have perceived in it features that correspond (at least partially) to their representation of that concept. However, because of this, the features that we then retrieve with our analyses might be incomplete or indirect reflections of the mental representation.

4.2 There are clear benefits and limitations of the current approach over, say, the reverse correlation studies. Some of these issues were brought up in the prior reviews and the authors have made some revisions. But I don’t think the authors are providing a balanced discussion of these issues – the authors really need to be moderating their

claims. On top of some of the limitations that have already been raised, I'm wondering about the impact of the training task and stimuli for the DNN. The DNN was trained on an object categorization task with the Imagenet stimuli – these stimuli are primarily close up views of objects with minimal background. Interestingly, the responses of observers in the current study to the DNN feature stimuli often contain more scene features (e.g. grass, sky). Would comparable results be obtained if the authors used a DNN trained on scene categorization? One feature of such trained networks is they often have more object-like features in earlier layers of the network. What impact would this have on the estimated internal representations? Further, one aspect of Imagenet that is well recognized is that there are a disproportionate number of some categories. Such categories include dogs and fish, which both feature prominently in the labels that observers gave for the DNN feature stimuli in the current study. Is it possible that the prominence of these labels in part reflects the nature of the Imagenet stimuli? The bottom line is that the choice of network, training task, and training stimuli are all factors that may affect the nature of the results reported here. Without explicitly testing their impact, it's hard to know how much the current results are influenced by them.

We agree with the reviewer that the specific DNN chosen and the way it was trained certainly influence the features that we sample. This in turn will likely influence our results. However, the largest impact will be on the categories that are labeled rather than the content of the visual representations uncovered. For example, dog and fish features might indeed be more prevalent in our stimuli because our network was trained on the ImageNet dataset, and this will increase the likelihood of participants labeling stimuli as dogs and fish. However, this will only happen if features are recognized as being dog and fish features by participants. Thus, it should mostly influence our ability to uncover with good resolution representations of these concepts, rather than bias their content in inaccurate directions. Nonetheless, it is possible that some bias occurs due to the specific network chosen: only some features among the set of all possible features are sampled and these would have been different with a different network. We now discuss these possibilities and further moderate our claims as suggested (see, e.g., lines 497-509):

Layers of other CNNs trained for object categorization (e.g., the more recent FixEfficientNet⁶¹ or Vision Transformer⁶²⁻⁶³) or of deep neural networks trained for other objectives, such as generative adversarial networks (GANs)^{24,64}, could also be used as the feature space. In addition, using a network trained for scene categorization might allow us to sample more complex multi-object features if we use a relatively high layer. Training the networks on datasets that more closely resemble the distribution of natural categories relevant to humans might further improve the results⁶⁵. Neural networks trained on biased datasets such as ImageNet (with an overrepresentation of dogs and other categories) are likely to bias the features that are sampled, in turn influencing the categories answered in an open labeling task such as ours. Although this does not alter the set of categories that can be reconstructed with our method (since any concept label recognized by the word embedding can be input), categories labeled more frequently might be reconstructed more successfully. In general, the specific architecture, training task, training stimuli, and learning rule are all factors that may alter the features sampled and influence our results.

4.3 The DNN feature stimuli look like texturized stimuli akin to things produced by the Simoncelli texture synthesis algorithm and used by Bria Long and Talia Konkle to investigate the contribution of mid-level features to the organization of human ventral visual cortex (Long et al, 2018, PNAS; Long et al, 2016, JEP:Gen; see also similar work by Jagadeesh and Gardner, 2022, PNAS). It's not clear to me that the current work provides that much more insight than any of these studies. Mid-level features are associated with high-level visual categories and if you ask observers which categories might be associated with stimuli showing different types of features they can tell you.

We thank the reviewer for raising these important works and for the opportunity to clarify the advance of our findings. Contrary to our study, participants in these studies were not asked to indicate in an open-ended manner what concepts they perceived in the stimuli. Moreover, the relationships between specific features and such categories were not analyzed. Our manuscript reports a novel method to uncover the features associated with multiple concepts, both at the group level and in individual observers, visualize them and analyze them in various ways. We now cite the related studies mentioned by the reviewer and note how our work builds on their findings (see lines 556-561):

Our work is also related to other studies assessing the contribution of mid-level features to the representations of high-level categories such as real-world size and animacy⁸⁰⁻⁸². However, these studies did not attempt to relate specific features with categories, nor did they target a large number of categories. This important distinction is what enables us to analyze and reconstruct the contents of visual representations on a large scale across conceptual space.

4.4 Abstract: "there is currently no general framework for providing direct access to representations of high-level visual concepts". True, but this manuscript does not achieve this. And what would be "direct access"?

By "direct access", we meant an ability to visualize the features that are part of the representations. However, we realize that this can be misinterpreted and have now modified our phrasing. We also rephrased the passage mentioning the absence of a general framework for investigating representations (lines 35-36):

However, there is currently no framework for reconstructing representations of multiple high-level visual concepts.

4.5 Throughout the manuscript, the authors tend to provide only qualitative overviews and summary statistics for the data. In many cases it's hard to understand exactly how the statistics were computed. I appreciate that the data will be made fully available, but I still think that more information should be provided in the manuscript or supplementary material. For example, the authors report that, "After corrections and removal of invalid responses, this corresponded to 2,578 unique words, of which 369 were provided at least 10 times each (Figure 2a). The most frequent words were "grass" (607 responses), "sky" (592), "tree" (450), "dog" (355) and "bird" (355)". It would be helpful to see a frequency distribution and a list of all 369 words. Similarly, on Lines 199-207 it would be helpful to see more data – how many different words per

reconstruction? How were the stats computed. These are just a couple of examples, but I think a fuller presentation of the data and greater clarity over the statistical testing throughout would strengthen the manuscript.

We have now modified our manuscript to include more details on the data and statistical tests throughout the main text (see., e.g., lines 157-162 and 215-219):

Although most responses were objects or concrete things, there were some more abstract concepts. Looking at the 350 validated labels (labels named 10 times or more and labels from the Visual Genome database; see below), four labels (1.1%) were not nouns (“green”, “white”, “black”, and “dark”) and an additional eight (2.3%) were clearly not basic-level (“animal”, “building”, “clothes”, “furniture”, “fabric”, “buildings”, “container”, and “vehicle”). All words were included in the analyses.

Despite the greater difficulty of this task, several concepts were recognized with high accuracy: the most common written label was the correct label for 37 of these concepts (significantly above the 1.1 concepts that would be labeled correctly on average by chance; tested with random permutations of participant responses across reconstructions, $p < .001$, one-sided; 95% C.I. = 28–47).

We also added a list of all words answered at least 10 times and their frequency distribution in the supplementary material (Figure S5 and Table S1) and we indicated the average number of unique words per reconstruction in the second validation (lines 214-215):

For each reconstruction, participants wrote many different labels. On average, there were 68.9 unique labels per concept (standard deviation across objects = 16.3).

4.6 Participants were asked to indicate “objects or concrete things” that they could see in the DNN feature stimuli. Why was it limited to three? On lines 134-135, it is stated that “we asked observers to indicate all categories they perceived in the stimuli”. This is inconsistent with the instructions described in the methods both in terms of the terms used and the fact that responses were apparently limited to three.

We limited the number of responses to 3 in order to control the duration of the experiment. We now specify this in the manuscript (lines 650-651):

[...] indicate the “objects or concrete things” that they saw, between one and three for each picture (to limit the duration of the experiment).

We also modified our description in lines 134-135 to be more accurate and in line with what is described in the methods:

[...] write any category (between one and three) they perceived in the stimuli [...]

4.7 The initial validation with a 2AFC is a weak test – I suggest that much less emphasis be placed on this experiment.

We now reduced the emphasis placed on the first validation in the Abstract and throughout the main text. See, as examples (abstract, lines 38-44, and discussion, lines 388-396):

This allowed us to reconstruct the mental representation of virtually any common visual concept, both those supplied by participants and other concepts extrapolated from the same semantic space. We successfully validated many of these reconstructions in separate participants. We further generalized the approach to predict behavior for new stimuli and in a new task, and to reconstruct representations of individual observers and of a neural network. This framework enables a large-scale investigation of conceptual representations.

Our method allowed us to successfully reconstruct the representations of several concepts. Seventy percent (270/350) were recognized above chance against other reconstructions in a 2AFC task and 37% (37/100) were accurately labeled in an open-ended labeling task. When analyzing the semantic content of labels, 85% (85/100) were well recognized, with semantically related responses being more frequent than semantically distant responses. We were able to recover the representations of diverse concepts, spanning multiple domains such as animals, vegetation, buildings, colors, materials, and objects. Importantly, it might be possible to successfully reconstruct additional concepts; we could only validate a finite subset of all possible representations.

4.8 Line 152: what's an invalid response?

Invalid responses are one-character words, numbers, words part of NLTK's (Loper & Bird, 2002) list of English stop words (i.e., words commonly used and without a lot of meaning, such as "the", "and", "I", etc.), and words unrecognized by the word embedding (see Methods: Visual-semantic matrix). We now specify this in the main text of the manuscript (lines 152-154):

After corrections and removal of invalid responses (single characters, numbers, stop words as defined by the NLTK Python library⁴⁶, and words unrecognized by the word embedding), this corresponded to 2,578 unique words [...]

4.9 Lines 322-324: "We reconstructed the network's representations in the same way as we did for the human representations. Resulting reconstructions often look superficially similar to reconstructions of human representations but with the concept rendered less clear or even unidentifiable (Figure 7a)." Why not run the same analyses as run for humans?

We thank the reviewer for the opportunity to clarify. We did use the same image synthesis algorithm to reconstruct the representations of the DNN and those of the human participants (see Methods: Reconstruction of mental representations). Then, to analyze similarities between DNN reconstructions and human reconstructions, we divided the data in halves for each "group" (DNN and humans) and we compared within-group correlations of the representations' features to between-group correlations, to obtain a measure of how unique the representations of both groups were. This analysis is similar to the one we used to analyze similarities between reconstructions of individual human observers (lines 357-379), except that here there are two groups instead of eight individuals.

REVIEWER #5

5.1 It would be helpful to clarify the scope of what is included as a “concept”. For example, most of the ones mentioned seem to be very concrete objects or “stuff”, but then there are colors like “white”? The concept “strange” is mentioned on line 424. Is “strange” or other adjectives like this included as a concept? What is the proportion of objects/stuff/adjectives within the set of concepts that you included? For objects, are the labels mostly basic-level (dog), or do they also include superordinate (animal) or subordinate (dog breed)?

Great question. We instructed participants to indicate “objects or concrete things” but ultimately did not restrict responses and included all words in the analysis (except one-character words, numbers, words unrecognized by the word embedding, and words part of NLTK’s list of English stop words; Loper & Bird, 2002). Although it is hard to quantify the number of concrete words unambiguously, an inspection of the 350 labels that were validated (those named 10x or more or present in the Visual Genome database) reveals that only four (1.1%) were not nouns (“green”, “white”, “black”, and “dark”; “strange” was very rarely named and so is not among those 350 validated labels). An additional eight (2.3%) were not basic-level (“animal”, “building”, “clothes”, “furniture”, “fabric”, “buildings”, “container”, and “vehicle”). We now include this information in the manuscript (lines 157-162):

Although most responses were objects or concrete things, there were some more abstract concepts. Looking at the 350 validated labels (labels named 10 times or more and labels from the Visual Genome database; see below), four labels (1.1%) were not nouns (“green”, “white”, “black”, and “dark”) and an additional eight (2.3%) were clearly not basic-level (“animal”, “building”, “clothes”, “furniture”, “fabric”, “buildings”, “container”, and “vehicle”). All words were included in the analyses.

5.2 Related to the above - in the 2AFC task where people are choosing between two labels for the reconstruction, is there ever a comparison between different types of things (like an object vs a color, or a superordinate category versus a basic-level category)? I can imagine this creating ambiguity, like if you have “animal” vs. “dog” for a dog image. How do you address this possibility?

In this task, we chose the non-matching label for each reconstruction randomly. Because the number of non-basic-level nouns was very small, this suggested kind of comparison occurred very rarely by chance. Moreover, participants were always asked to choose the *most* appropriate label, which in the reviewer’s example would be “dog”. We have clarified that the non-matching label was chosen randomly and that nearly all comparisons were between basic-level categories because there were so few labels at other types/levels (lines 197-199):

Note that because the non-matching label was chosen randomly and that there were few non-basic-level category labels, nearly all comparisons were between two basic-level category labels.

5.3 For the result described on lines 208 – 214: “Among the large number of concepts that could be reconstructed, it was still possible to distinguish between closely related and visually similar concepts ... singular and plural forms of the same concept could also be differentiated” Please clarify what is meant by “differentiated” here. Is this just based on visual inspection of the images? Or is there an experimental validation/quantification of this claim? This observation is also repeated in the discussion lines 366 – 369. If it is just based on visual inspection, this claim is misleading and should be modified.

This was an observation based on visual inspection. We have now removed this passage in the discussion and we nuanced our claim in the Results section (see lines 229-236):

It was possible to reconstruct many closely related concepts. A visual inspection shows that subtle differences in visual features between these can apparently still be revealed, although additional validations would be necessary to conclude this with certainty (Figure 4). This also seems to be true for singular and plural forms of the same concept, with more smaller repetitions of the concept across the image when the plural form of a given word was input (Figure 4f). Note that 270 does not seem to be the upper bound of the number of concepts that can be reconstructed: other words can be input that result in seemingly successful reconstructions (e.g., pond, bushes and shark on Figure 4) although, again, we cannot know for sure without validating them too.

5.4 It would be worth discussing the possible limitations related to the dataset that the model is trained on (ImageNet). For example, it looks like the category “dog” shows up quite a lot in the responses (Figure 2 C-D), more so than other animals. This might be related to the fact that there are a lot of dog breeds in ImageNet. Does the set of concepts that you can reconstruct depend on what categories are present during model training?

The specific DNN chosen and the way it was trained influence the features that we sample. This in turn likely influences the categories that are labeled. The set of categories that can be reconstructed does not depend on the labels given, since any concept recognized by the word embedding can be reconstructed with our method, but categories labeled more frequently may result in more successful reconstructions. We now discuss these possible limitations in the discussion (lines 501-509).

Training the networks on datasets that more closely resemble the distribution of natural categories relevant to humans might further improve the results⁶⁵. Neural networks trained on biased datasets such as ImageNet (with an overrepresentation of dogs and other categories) are likely to bias the features that are sampled, in turn influencing the categories answered in an open labeling task such as ours. Although this does not alter the set of categories that can be reconstructed with our method (since any concept label recognized by the word embedding can be input), categories labeled more frequently might be reconstructed more successfully. In general, the specific architecture, training task, training stimuli, and learning rule are all factors that may alter the features sampled and influence our results.

5.5 In the introduction, line 101-102: “An additional drawback of the reverse correlation and brain-based reconstruction approaches is that they focus on small sets of

experimenter-defined stimuli.” This doesn’t seem true – some of the brain-based approaches that you cited do model a continuous stimulus space, so they can reconstruct stimuli that are not in the training set, not just a fixed set of stimuli. Same for the “superstimulus” approaches (Bashivan 2019, etc). Please be more precise with this statement and what is the novelty here.

We agree that this sentence was inaccurate, the drawback was mainly directed towards reverse correlation studies. We have now modified our claims (lines 101-102):

An additional drawback of reverse correlation studies is that they focus on reconstructing a limited number of predefined stimuli⁸⁻¹¹. [...]

5.6 For the first experiment, why did you average the feature vectors for all reported words together (lines 644 – 645) instead of keeping them separate? I am wondering if this would obscure any information, especially if the reported words for an image are very different. It looks like in some cases the random images have multiple distinct “things” in them, in different regions of the image, as opposed to just one concept. When people reported multiple concepts, were they usually very similar, or were they different? What if different people reported very different concepts? Please provide some more justification for this point.

Each image was shown only once, to only one observer. We only average the vectors of the different labels given by this observer, if more than one is given. This was done for statistical reasons, to avoid repeating the visual features multiple times (once for every label) in the matrix of independent variables prior to the creation of the visual-semantic matrix and thus preserve the random nature of their sampling. In addition, we wanted to avoid giving more weight to trials with more responses or to individuals who wrote more labels. Moreover, averaging the semantic features of the labels in response to one image should not affect the results (beyond avoiding the biases mentioned above) because these are associated to the same visual features and our outer product is equivalent to averaging visual features weighted by semantic features. (Labels written for a given stimulus were indeed often very different, targeting different parts of the image, but there is no way to know which parts are related to which labels with our paradigm.) We now provide more justification for this point (lines 693-697):

For each stimulus (shown only once to one observer), the vectors of semantic feature values of all reported words were then averaged into one vector, to give equal weights to all trials regardless of the number of responses and to preserve the random sampling of visual features by not repeating the same visual features for multiple words in the creation of our visual-semantic matrix.

5.7 How does the distribution of the “target” CNN activation vectors (line 678) differ from the distribution of the “random” CNN vectors used for the CNN-noise stimuli? I see that for the “random” CNN feature vectors, they are forced to have the distribution/covariance as the activations for real images. Do the “target” feature vectors for the reconstructions also have this property, or are they allowed to be less natural? From a glance at Figure 2, it looks like maybe the reconstructions are less

natural in appearance than the CNN-noise images (maybe more high-freq artifacts?). Are there any consistent visual differences across these stimulus types?

For the CNN-noise stimuli, the CNN feature values were randomly drawn from the distributions of possible values in natural images. For the reconstructions, no such distributions were created. The target feature values were obtained purely from the visual-semantic matrix (that is, they are the product of the label's semantic values with our visual-semantic matrix). As such, the values can indeed be outside the distribution of values in natural images.

There may be visual differences between reconstructions and stimuli beyond the fact that one is optimized to represent a specific concept while the other is not (note that on Figure 2, images depicted are not exactly reconstructions of concepts as they are maximizing the value of a specific visual feature). However, it is hard to say with certainty. As examples, you can see on Figure 5a examples of reconstructions but using null visual-semantic matrices (the same as normal reconstructions but using inaccurate matrices). These look quite similar to stimuli such as those visible in Figures 1 and 2, despite each perhaps having a smaller number of visible visual features.

We now clarify these facts in the Methods (lines 729-730 and 751-752):

Note that these CNN feature values can be outside the distribution of values in natural images.

Because of these changes, there may be small baseline visual differences between reconstructions and stimuli.

5.8 I found the paragraph from lines 183 – 198 a bit confusing in how the different sets of images are presented. For example you have: “the 250 words most-named in the experiment and the top 100 most frequent words from the Visual Genome database labeled less than 10 times by participants.” But then, some of the 250 words in the first group are also part of visual genome? I got confused when you describe the “209” concepts, and then later the “47” concepts that are rarely or never named. Please clarify these different groups in the text.

We thank the reviewer for the opportunity to clarify. We have now improved our descriptions of the different sets of images that were validated (see lines 192-209):

We validated the reconstructed representations of 350 words in an additional behavioral experiment: the 250 words most-named in the experiment plus the top 100 most frequent words from the Visual Genome database⁴⁸ labeled less than 10 times by participants (and thus not part of the 250 most-named). On each trial, 50 participants were shown a reconstruction and had to choose between two labels: the true label and an incorrect label randomly chosen among the labels of the other reconstructions²⁴. Note that because the non-matching label was chosen randomly and that there were few non-basic-level category labels, nearly all comparisons were between two basic-level category labels. The reconstructions for the 250 most-named words were recognized well on average, with a mean accuracy of 88% (significantly above chance, $p < .001$). For the 100 additional frequent concepts

from the Visual Genome database, the mean accuracy was 74% ($p < .001$). When considering all validated labels in the Visual Genome database, irrespective of whether they were named more or less than 10 times during the experiment (209 concepts), the mean accuracy was 84% ($p < .001$). Overall, 270 concepts (out of the 350 that were validated) were individually recognized significantly above chance (accuracy $> 75\%$; $d_z > 1.73$; Figure 3c). This proportion of significant concepts was higher than would be expected by chance (random resampling of participants to create an empirical null distribution, $p < .05$, one-sided, FWER-corrected). Within the 100 concepts named less than 10 times by participants, 47 were individually significant.

5.9 For the second “validation” task described on lines 199 – 207, it seems like the accuracy is relatively low, even though it is above chance. I wonder if there is another way to assess accuracy here, as opposed to looking for an exact match between the written and actual concept, which is quite hard. For example could you compute how aligned are the embeddings of the written and actual concepts? This would capture if the guessed concept is “close” to the real concept.

Based on the reviewer’s suggestion, we performed additional analyses of this validation data using semantic distances. We now report results indicating that for 85 of the 100 validated concepts used in this study, semantically close responses were significantly more frequent than semantically far responses (significant linear fit, $p < .05$, FWER-corrected). See lines 222-228:

[...] This test relies on writing the exact correct label, however. To account for participants writing inexact but closely related labels, we analyzed the semantic features of the responses and how similar they were to those of the true labels. Strikingly, we found that responses that were close to the true label in semantic space were more frequent than responses that were far in semantic space for 85 of the 100 validated concepts (inverse linear relationship between response semantic distance and frequency, significance test of the slope coefficients, $p < .05$, one-sided, FWER-corrected).

See also the Methods section, lines 976-981:

We then analyzed how many concepts were generally well labeled with semantically close responses, even if these were not the concept’s exact true label. To do so, for each concept, we fit a first-degree polynomial equation between the semantic distance of each unique response to the true label and its frequency. We then transformed the slope coefficient into a t value and computed its significance using the Student’s t distribution, Bonferroni-correcting for the 100 statistical tests.

5.10 Clarification on line 755: “We binarized this vector of correlation coefficients so that the number of stimuli predicted as containing the concept matched the number of stimuli containing it. “I couldn’t figure out what this means exactly, did you threshold the correlation coefficients to make them 0/1? Why was this done? Please clarify this method.

We apologize for the confusion. Yes, we thresholded the correlation values so that they were either 0 or 1. This was done so that we could obtain a binary prediction (based on CNN feature values) of whether the concept was present in the image or not and compare these predictions to the ground truth binary values indicating which stimuli were associated with

the concept by participants at least once. We have now clarified our method and its purpose (lines 803-812):

To predict the stimuli associated with a specific concept, we used the visual-semantic matrix from the main experiment and the set of stimuli and responses from the Individual Representations experiment. For each of the 10 most-named concepts (grass, sky, tree, dog, bird, water, animal, snake, building, eyes), we created a Boolean vector of 0s and 1s indicating which stimuli were associated with the concept at least once. Then, we created another Boolean vector representing our predictions of which stimuli were associated with the concept solely based on its visual features. To do so, we computed how much the CNN feature values of each stimulus correlated to the CNN feature values associated with the concept (as obtained with our visual-semantic matrix) and thresholded this vector of correlation coefficients so that the number of stimuli predicted as containing the concept matched the number of stimuli containing it. [...]

5.11 Figure 3 C is referenced in the text before A and B, these should be referenced in order.

We thank the reviewer for catching this. We have now modified the text so that figure panels are mentioned in order.

REVIEWER #6

6.1 CNN feature values were more correlated within group" -- it would be helpful to have more clarification here. Is the comparison being shown here CNN within group vs. CNN-human across group, or is it CNN within group vs. human within group corrected by the across-group measure? It would be nice to see the comparison across all 3 measures (within CNN, within human, across CNN-human), maybe even as a small 3x3 correlation matrix in the figure.

By “within group”, we meant both within the CNN and within humans, compared to across groups (CNN vs. human). We now mention in the main text all correlation values (across groups = 0.49 and 0.46 for both sets of trials; within model = 0.85; within humans = 0.56). Moreover, to avoid confusion between the “CNN feature values” (which are the visual features of the representations, associated with both humans and the deep neural network) and the deep neural network itself, we simplified the phrasing, use “visual features” in lieu of “CNN feature values” in this section, and use the phrase “deep neural network” in lieu of “convolutional neural network” to denote the network in this section (lines 334-355). We made similar modifications and added an explanatory note in the related methods section.

Investigating the representations of the deep neural network

We then aimed to determine whether human representations were different from the representations of the neural network used for image synthesis. This is important for several reasons. First, uncovering differences between the representations of humans and of a deep neural network (DNN) would show the added value of our method and that using a DNN’s representations as a proxy for human

representations is insufficient. Moreover, it would reveal that our method is a useful tool to analyze representations of artificial neural networks, in addition to human representations, and potentially to compare representations of different DNNs to each other (to identify the ones that capture behavior better). To achieve this, we repeated the experiment but used as the responses for each stimulus the labels of the three classes (out of the 1,000 ImageNet classes) that the network estimated had the highest probability of being depicted in that stimulus. We reconstructed the network's representations in the same way as we did for the human representations. Resulting reconstructions often look superficially similar to reconstructions of human representations but with the concept made less clear or even unidentifiable (Figure 7a). We then analyzed whether representations were significantly different between groups (humans vs. DNN) while accounting for different noise levels. Specifically, we projected the visual-semantic matrices of both humans (Figure 7b) and the network (Figure 7c) to a common semantic space, divided the data in halves, and compared the features of representations within and across groups. Visual features were more correlated within group ($r_{\text{within, DNN}} = 0.85$; $r_{\text{within, human}} = 0.56$; $r_{\text{between}} = 0.49$ and 0.46 ; $r_{\text{within, mean}} = 0.70$ vs $r_{\text{between, mean}} = 0.48$; 95% C.I. = 0.701 – 0.705 vs 0.477 – 0.483 ; $Z = 8.73$; $p < .002$, two-sided), suggesting limits in the correspondence between DNNs and humans (see also, e.g., ref.⁵¹).

REVIEWERS' COMMENTS

Reviewer #4 (Remarks to the Author):

The authors have fully addressed my concerns and made appropriate revisions to the manuscript. I have no further comments.

Reviewer #5 (Remarks to the Author):

The authors have addressed all of my comments from the review, and I would recommend this paper for publication.

One small thing I noticed on line 215: “standard deviation across objects” – is this supposed to say “concepts” or “subjects”?

Reviewer #6 (Remarks to the Author):

I only had a minor comment in the previous review and the authors did a great job at addressing it clearly. I have no further comments and am happy with the manuscript in its current form.

REVIEWER #4:

The authors have fully addressed my concerns and made appropriate revisions to the manuscript. I have no further comments.

We thank the reviewer for their comments, our manuscript was much improved as a result of the review process.

REVIEWER #5:

The authors have addressed all of my comments from the review, and I would recommend this paper for publication.

One small thing I noticed on line 215: “standard deviation across objects” – is this supposed to say “concepts” or “subjects”?

We thank the reviewer for their comments, our manuscript was much improved as a result of the review process. We have now modified this passage, “objects” should have been “concepts”.

REVIEWER #6:

I only had a minor comment in the previous review and the authors did a great job at addressing it clearly. I have no further comments and am happy with the manuscript in its current form.

We thank the reviewer for their comments, our manuscript was much improved as a result of the review process.